# MemER: Scaling Up Memory for Robot Control via Experience Retrieval

**Ajay Sridhar**[*]   **Jennifer Pan**[*]   **Satvik Sharma**   **Chelsea Finn**
Stanford University
{ajaysri, jrpan}@stanford.edu

## Abstract

Humans rely on memory to perform tasks; our goal is to endow robot policies with the same ability. Naively conditioning on long observation histories is computationally expensive and brittle under covariate shift, while indiscriminate subsampling of history leads to irrelevant or redundant information. We propose a hierarchical policy framework, where the high-level policy is trained to select and track previous task-relevant keyframes from its experience. The high-level policy uses selected keyframes and the most recent frames when generating text instructions for a low-level policy to execute. This design is compatible with existing vision-language-action (VLA) models and enables the system to efficiently reason over long-horizon dependencies. In our experiments, we fine-tune Qwen2.5-VL-7B-Instruct and $\pi_{0.5}$ as the high-level and low-level policies respectively, using demonstrations supplemented with minimal language annotations. Our approach, MemER, outperforms prior methods on three real-world long-horizon robotic manipulation tasks that require minutes of memory. Videos and code can be found at https://jen-pan.github.io/memer/.

## 1 Introduction

In recent times, we have seen significant strides in the language-following and generalization capabilities of robotic manipulation policies (Brohan et al. (2023); Intelligence et al. (2025); Kim et al. (2024); NVIDIA et al. (2025)). While these policies are getting better for real-world deployment, a critical limitation remains: the absence of long-term memory. Memory allows humans to handle the inherent partial observability found in their environment. For instance, if a person wanted to make a sandwich, they would have to recall where they saw the jar of peanut butter or the knife, especially if these items were not recently viewed. The ability to form and retrieve long-term visual memories is a crucial step towards robots solving complex, multi-step tasks. The goal of this paper is to provide an effective way to enable existing generalist policies to solve tasks that require long-term visual memory.

Because conditioning on long sequences of high-dimensional image and video sequences is computationally expensive, many existing generalist end-to-end policies are trained with little to no visual history. The high memory cost makes training prohibitively expensive and model deployment unusably slow. Furthermore, long observation histories can often introduce a form of overfitting — shortcut reliance on spurious correlations between inputs and demonstrator actions (Torne et al., 2025). The policy misgeneralizes under its own state distribution, leading to performance degradation during deployment due to compounding covariate shift between states visited by the demonstrator policy and the learned policy. The suboptimal policy will generate histories that differ from those seen by the expert, which is only made worse as observation history lengthens.

Some past works have shown it is possible to expand the observation context of their policy via auxiliary losses (Torne et al., 2025), or by finetuning pretrained foundation models for action prediction with native memory capabilities (Fang et al., 2025). Although these methods significantly increase the types of tasks a robot can execute, they are challenging to naively scale to long histories. To overcome this, policies must learn to filter out the task-relevant information from the full

---

[*]Equal contribution

historical context to prevent the memory footprint from exploding on tasks that require long-range dependencies.

To this end, we propose approaching long-term memory for robotic policies with a hierarchical framework. The high-level policy is a fine-tuned video-understanding VLM trained to output action subtasks and, most importantly, to select keyframes from its fixed recent context that represent important information it will need to remember to solve the task. The low-level policy is a generalist robot policy fine-tuned to execute the subtasks specified by the high-level policy. Together, the low-level policy handles the robot-specific challenges of the task that require high-frequency inference such as kinematic control, and the high-level policy manages planning and memory-specific aspects of the task such as deciding what object or tool to manipulate next based on the high-level task and its memory. We take advantage of the fact that these open-source VLMs are finetuned on large amounts of video understanding data. With this strong prior, we find that we only need a relatively small number of teleoperated robot demonstrations with additional annotations to adapt these VLMs to accomplish robot-specific memory-based tasks (Bai et al., 2025).

Our contribution is MemER, a framework for scaling up **Mem**ory in robotic control via **E**xperience **R**etrieval. We demonstrate MemER's ability to utilize task-relevant past information effectively on three complex long-horizon tasks that require up to a couple of minutes of memory. To the best of our knowledge, our real-world robotic tasks necessitate reasoning over more image observations than prior work.

## 2 RELATED WORK

**Memory in Robotics and Long-Context Policies.** Memory is essential for generalist robots to complete complex tasks. Prior work primarily studies memory in the context of comparatively short-horizon tasks. For example, Torne et al. (2025) and Fang et al. (2025) use different approaches to extend the context of imitation policies from a few frames to at most two dozen. Our work investigate tasks that require building memory from hundreds of frames. Unlike previous approaches, our method can choose to include task-relevant frames in the context spanning the entire episode. Another body of work investigates the compression of images in the policy's context. Li et al. (2025a) compresses similar observations in pixel space, which is effective for stationary camera setups but struggles with wrist-mounted cameras that are necessary for most dexterous manipulation tasks. Memory has also played a major role in robotic navigation research. Some navigation works represent memory with an explicit geometric and/or semantic map of the environment Henry et al. (2012); Yu et al. (2024). However, spatial maps of the environment are hard to apply in manipulation tasks since the robot often modifies the environment. Other works directly prompt API-based VLMs with video context to decide where the robot should navigate (Chiang et al., 2024; Sharma et al., 2023; Chen et al., 2024). We found that existing API-based VLMs are not sufficient to reason about robot affordances for our long-horizon, memory-aware tasks (Table 2), so we resort to finetuning open-weight models.

**Foundation Models and Long-Horizon Tasks in Robotics.** Recent progress in vision-language-action models (VLAs) have allowed for impressive generality in robotics. VLAs combine web-scale pretraining with expressive action decoding mechanisms to execute real-world tasks. Conceptually, two paradigms have emerged. The first is a single end-to-end model that directly maps images and a language task to actions (Intelligence et al., 2025; Brohan et al., 2023; Fan et al., 2025). The second is a hierarchical approach that uses a high-level policy to output an intermediate representation to guide a low-level policy. Possible intermediate representations include latent embeddings (Shentu et al., 2025; Wen et al., 2025; Wu et al., 2024), language subtasks (Shi et al., 2025; 2024), and waypoints (Li et al., 2025b). Prior work has shown that hierarchical approaches improve performance on long-horizon tasks (Shi et al., 2025; Wu et al., 2024; Wen et al., 2025) by introducing a temporal abstraction: a high-level policy operates at a lower frequency and decomposes a complex task into simpler subgoals (Zhang et al., 2023). The low-level policy can then focus on high-frequency, reactive motor control to achieve the current subgoal. Other methods that integrate LLM-based high-level planning with task and motion planning, imitation learning, and RL (Dalal et al., 2024; Zhou et al., 2024) similarly tackle long-horizon manipulation via task decompositions, but rely on a well-specified planning stack and do not provide an explicit long-term memory mechanism over raw visual histories.

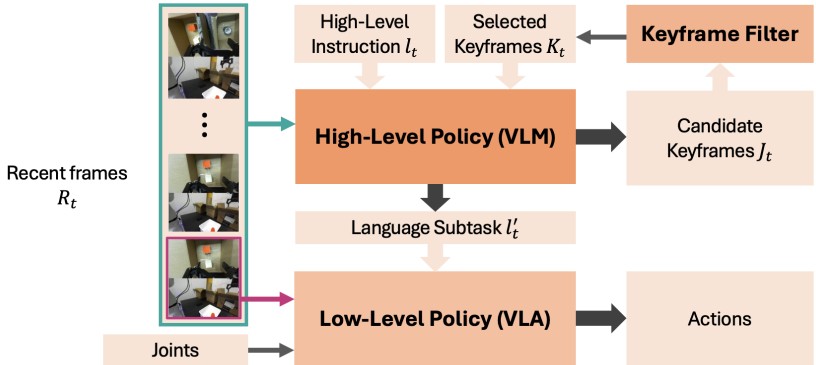

Figure 1: **Architecture of MemER.** The high-level policy processes task instructions, selected keyframes (if any), and recent images from base and wrist-mounted cameras to generate low-level language subtasks and candidate keyframes (if any). The low-level policy uses the subtask, current image, and robot joint states to produce actions. The candidate keyframe(s) are processed by the keyframe filter to obtain the selected keyframes for input during the next step of inference.

Our work builds on the second paradigm, using language subtasks as the intermediate representation. What distinguishes MemER from prior hierarchical approaches is the addition of a stable, persistent memory mechanism that preserves salient long-range dependencies while keeping inference fast enough for real-world deployment. We show that this memory system is essential for complex, long-horizon tasks that span multiple minutes, on which standard hierarchical methods with little or no memory reliably fail.

**Video Keyframe Selection.** Outside of robotics, previous work in computer vision has also studied incorporating longer contexts for VLMs (Goletto et al. (2024); Manigrasso et al. (2025)). Similar to our work, other works have used keyframe selection to improve video understanding and question answering (Yu et al., 2023; Ranasinghe et al., 2025). Many such methods incur a high per-frame cost because they estimate frame importance via separate multimodal-LLM calls. These methods are not directly applicable for robotic tasks because with increasing video context lengths they would not meet the task's latency constraint during inference. Hu et al. (2025a) uses lightweight models to score all frames in a single pass, which reduces per-frame cost but lacks the ability to continuously stream image observations. Departing from existing VLM work for VQA, Hu et al. (2025a) uses non-uniform frame sampling through a lightweight scoring model; in contrast, we achieve non-uniform sampling without additional models. Designed for real-world robotics, our method emphasizes low-cost inference and streaming support.

## 3 MEMER

### 3.1 PRELIMINARIES

**Language-Conditioned Control Policies.** Language-conditioned robot policies are typically trained to model the conditional distribution $p(\boldsymbol{A_t}|o_t)$, where $\boldsymbol{A_t} = [a_t, a_{t+1}, \ldots a_{t+H-1}]$ is a chunk of actions modeled from the current timestep $t$ to $H$ timesteps in the future (Zhao et al., 2023) and $o_t$ is the robot's current sensor observation. The current observation is usually formulated $o_t = [\boldsymbol{I_t}, l_t, \boldsymbol{q_t}]$, where $\boldsymbol{I_t} = [I_t^1, I_t^2, \ldots, I_t^n]$ are images from multiple cameras, $l_t$ is the language instruction, and $\boldsymbol{q_t}$ are the proprioceptive inputs from the robot (i.e. joint angles and gripper state) (Black et al., 2024; Team et al., 2024).

**Memory-Based Tasks.** We consider a set of tasks such that robot policy must leverage past information to successfully complete them due to partial observability in the environment. In other words, a robot policy trained to model $\pi(\boldsymbol{A_t}|o_t)$ could not complete the task, but a policy trained to model $\pi(\boldsymbol{A_t}|o_{0:t})$ could.

**Hierarchical Policies.** In order to execute complex, long-horizon tasks, we follow Shi et al. (2025) and hierarchically decompose the robot policy into a low-level control policy ($\pi_l$) and a high-level

policy ($\pi_h$) that generates instructions.

$$\pi(\boldsymbol{A_t}|o_t) = \pi_l(\boldsymbol{A_t}|[\boldsymbol{I_t}, l_t', \boldsymbol{q_t}])\pi_h(l_t'|\boldsymbol{I_t}, l_t) \tag{1}$$

The high-level policy models the conditional distribution $\pi_h(l_t'|\boldsymbol{I_t}, l_t)$, where $l_t'$ is the current language subtask, which we also refer to as subtasks, that the low-level policy conditions on to complete the overall instruction $l_t$. The high-level policy could be represented as a separate VLM Shi et al. (2025) or share the same weights as the low-level control policy (Intelligence et al., 2025). In our method, the high-level policy will be responsible for reasoning about memory.

**Data Collection** Similar to prior work, we use language subtasks $l_t'$ to label each observation in a trajectory (Shi et al., 2024; 2025; Intelligence et al., 2025). We end up with a dataset of trajectories of the following tuple $(\boldsymbol{I_t}, \boldsymbol{q_t}, l_t', a_t)$ to train our high-level and low-level policies. For example, we can take the task "search for ketchup" and break it down into the following subtasks: "look in left bin", "look in right bin", and "take out ketchup from right bin." Examples of the subtasks and ground-truth frames for each task are shown in the Figure 3. In our expert data-collection setup, the operator executes a prescribed primitive and presses a key upon completion to advance to the next primitive. Consistent with previous work, we supplement the low-level policy training set with 10–15 intervention demonstrations to improve robustness at deployment Hu et al. (2025b); Kim et al. (2025). To collect the intervention data, we first initialize the trajectories in common failure states we have seen the low-level policy reach, then we teleoperate the robot back into an in-distribution state. Since the low-level policy only uses the current frame, we can have the robot start from bad states then demonstrate the correct behavior from those states.

## 3.2 High-Level Policy

Our method builds on the common hierarchical VLA paradigm, as in Shi et al. (2025), and extends it with the ability to tackle long-horizon tasks that require memory. We choose to adopt a hierarchical apporach because open source VLMs like Qwen2.5-VL-7B-Instruct have a strong video understanding prior from the video datasets they have been trained on, and thus can be adapted for memory-based planning (Bai et al., 2025). We use a finetuned VLM as the high-level policy to both nominate candidate keyframes and predict subtasks for the low-level policy during closed-loop control, as shown in Figure 1. The candidate keyframes are then filtered for redundancy and added to a group of selected keyframes that the high-level policy conditions on continuously when predicting the next subtask and candidate keyframes. Concretely, at each timestep, we feed our high-level memory policy 1) the last $N$ frames per camera $\boldsymbol{R_t} = \boldsymbol{I_{t-N+1:t}}$, where $N$ is the integer context-window shared across cameras, 2) the high-level task instruction $l_t$, and 3) previously selected keyframes $\boldsymbol{K_t} \subseteq \boldsymbol{I_{0:t-N+1}}$, where practically $|K_t| \leq 8$. The high-level policy then predicts two things: (i) the current subtask to execute $l_t'$ and (ii) the candidate keyframes $\boldsymbol{J_t} \subseteq \boldsymbol{R_t}$, a subset of frames from the recent context. All together, our high-level policy models $\pi_h(l_t', \boldsymbol{J_t}|\boldsymbol{R_t}, \boldsymbol{K_t})$. The low-level policy conditions on $l_t'$ to predict the direct joint velocities for the robot as described in Section 3.3. In parallel, $\boldsymbol{K_{t+1}}$, the selected keyframes for the next timestep of high-level inference, are calculated from the sequence of all candidate keyframes predicted since the start of the trajectory $\boldsymbol{J_{0:t}'} = (\boldsymbol{J_0}, \boldsymbol{J_1}, \ldots, \boldsymbol{J_t})$ using a simple 1D single-linkage clustering algorithm described in the following paragraphs.

**Building Visual Memory.** To build visual memory, our keyframe filter consolidates observations that have just exited our recent context, $\boldsymbol{R_t}$, into our selected keyframes $\boldsymbol{K_t}$. The filtering process operates on the temporal indices of these keyframes. The approach is task-agnostic and allows us to have coverage of all frames in the stream. By timestep $t$, the high-level policy has given us $\boldsymbol{J_{0:t}'} = (\boldsymbol{J_i})_{i=0}^{t}$, the sequence of candidate sets nominated up to timestep $t$. We begin by extracting the temporal index of every frame nominated in this sequence and pooling them into a single temporally ordered list $\boldsymbol{G_{0:t}}$. Importantly, we preserve duplicate indices, as this allows the subsequent median selection to aggregate repeated nominations and yield the most representative frame from each cluster.

We then create clusters, $\boldsymbol{C_i}$, for all candidate indices in $\boldsymbol{G_{0:t}}$ by grouping keyframe indices that are at most $d$ apart from each other. For example, if $\boldsymbol{G_{0:t}}$ has temporal indices $\{1, 3, 3, 4, 10\}$ and $d = 5$, then we would have two clusters $\boldsymbol{C_1} = \{1, 3, 3, 4\}$ and $\boldsymbol{C_2} = \{10\}$. After constructing the sequence of clusters, $(\boldsymbol{C_1}, \boldsymbol{C_2}, \ldots)$, we select the median index from each $\boldsymbol{C_i}$ to be that cluster's representative keyframe. The keyframes corresponding to these final median indices represent $\boldsymbol{K_t}$,

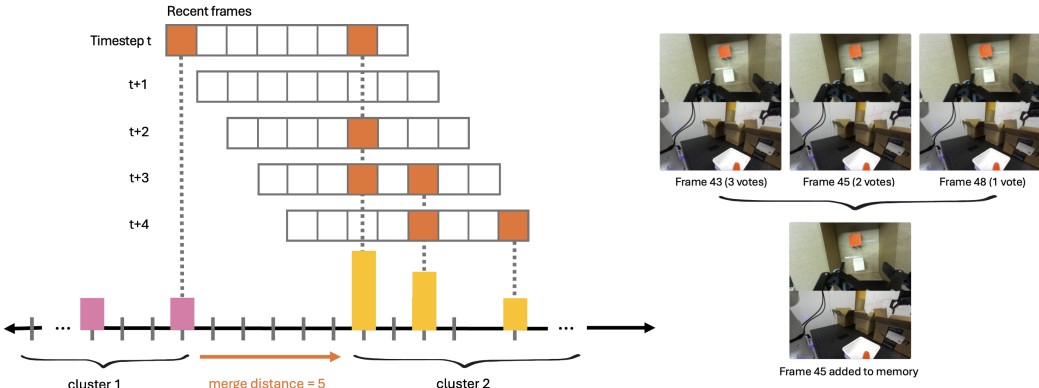

Figure 2: **1D single-linkage over nominated frames.** At each timestep, the high-level policy nominates candidate keyframe(s), as highlighted in orange. All candidate keyframes are aggregated across time with 1D single-linkage using a merge distance of $d = 5$ frames, yielding disjoint clusters. For each cluster, the colored bars indicate nominations for the observation at that timestamp, with bar height proportional to the number of nominations received. We select one representative frame per cluster by taking the median keyframe of all the candidates, and add that frame to memory.

the selected keyframes for timestep $t$. For efficiency, clusters that have indices less than $t - N + 1 - d$ do not need to be recalculated. Figure 2 is a visualization of the clustering and keyframe selection during deployment.

## 3.3 PRACTICAL IMPLEMENTATION OF MEMER

**Training the Low-Level Policy.** For our low-level robot policy, we finetune a version of $\pi_{0.5}$ Intelligence et al. (2025) trained on the DROID dataset Khazatsky et al. (2025). Given we have trajectories of the tuple $(\boldsymbol{I_t}, \boldsymbol{q_t}, l_t', a_t)$, we can train our low-level policy to model the conditional distribution $\pi_l(\boldsymbol{A_t}|\boldsymbol{I_t}, \boldsymbol{q_t}, l_t')$. We choose to finetune the $\pi_{0.5}$ checkpoint trained on the DROID dataset due to its strong out-of-the-box behavior on the DROID setup we use to conduct all of our experiments. Consequently, we find that we need only 50 demos of long-horizon trajectories to finetune a strong low-level policy. We finetune a single low-level policy across all three tasks. Refer to Appendix A for the specific training parameters.

**Training the High-Level Policy.** For our high-level policy, we finetune Qwen2.5-VL-7B-Instruct to predict two things: 1) the current subtask to execute and 2) any task-relevant keyframes to remember from the most recent frames (as described in Section 3.2). We finetune a single high-level policy on all three tasks, and we observe this gives the added benefit of stronger object generalization (see Appendix F for comparisons with the single-task variant of MemER). We freeze the weights of the vision encoder and projection layer during finetuning for training efficiency and to preserve the visual prior.

**Annotating Keyframes for the High-Level Policy.** To label keyframes for each task, we employ a semi-automatic annotation procedure. First, we extract the boundary frames between consecutive subtasks, which are simply the last frame of each subtask segment. Next, we review a small number of demonstrations (~3) to determine a simple annotation rule per subtask—deciding whether or not to keep the last frame of that subtask segment as a ground-truth keyframe, since these transition points usually contain a visually informative state. For example, the rule may indicate selecting the last frame in "look inside the center bin," or no frame for "reset scooper position." Once established, these rules are fixed per subtask and automatically applied to all demonstrations of each task; this process is not a manual, per-frame effort, but a quick, one-time setup that makes keyframe labeling practically free. The resulting set of keyframes forms the ground-truth targets used to train the high-level policy. See Appendix E for the specific keyframe annotation rules for all of the subtasks.

**Closed-Loop Deployment.** Our policy decomposition is the following:

$$\pi(\boldsymbol{A_t}|o_{0:t}) = \pi_l(\boldsymbol{A_t}|\boldsymbol{I_t}, \boldsymbol{q_t}, l'_t)\pi_h(l'_t, \boldsymbol{J_t}|\boldsymbol{I_{t-N+1:t}}, \boldsymbol{K_t}) \tag{2}$$

The interaction between the low-level and high-level policy for closed loop deployment is shown in Figure 1. The low-level policy predicts $\pi_l$ actions chunks at $\sim$2Hz, while the the high-level policy $\pi_h$ predicts keyframes and subtasks at roughly $\sim$1Hz. We run both policies on their own server. Like Shi et al. (2025), we choose to run the policies asynchronously, as we find it to improve performance. While the high-level policy is predicting the next primitive, the low-level policy conditions on the latest predicted primitive. We add the image observations sampled at 2Hz to a queue, and then send this queue to the high-level policy to query the next primitive prediction after the current high-level policy prediction is complete. Following Anonymous (2025), we found that linearly interpolating the weights of the finetuned high-level policy with its base model Qwen2.5-VL-Instruct-7B improves performance of the hierarchical policy on most tasks (more details in Appendix H).

## 4 EXPERIMENTS

In this section, we aim to evaluate the extent to which our method and alternative approaches can tackle long-horizon manipulation tasks that require some form of memory. We first describe our tasks and evaluation protocols, then we discuss the following questions:

1. To what extent can our approach tackle tasks that require memory, in comparison to a memory-less policy (i.e. current robot foundation models), a human high-level (Human HL) policy, and other naive approaches?

2. How does our high-level policy, fine-tuned from an open-source VLM, compare to proprietary off-the-shelf vision-language models?

3. How does representing memory via images compare to other modalities?

We design three complex, real-robot tasks that entail using memory in multiple distinct ways, including remembering object locations, keeping track of previously completed actions, and counting repeated task steps, as illustrated in Fig. 3. Since all of the tasks are long-horizon, we record different metrics for each task to provide a granular view of task completion.

**Object Search.** In the task, we randomly place three to five objects across three opaque bins. Then, the robot is sequentially given three objects to find; each new instruction is issued only after the robot has attempted to retrieve the previous object. Our goal is an optimized search: the robot remembers which bins it has already examined (and what it saw), skips re-searching them, and explores additional bins only as needed; it should proceed directly to the target bin if it has already been searched. This task requires cross-episodic memory as finding each object is its own $l_t$, thus requiring recall of information gathered while executing prior instructions. We train and test with the same set of 15 objects, which are various small toys. *Evaluation metric.* We measure task completion by two criteria for each of the three objects: successful retrieval and adherence to the optimal path without unnecessary exploration, for a maximum score of 6 (2 points per object).

**Counting Scoops.** In this task, the robot is asked to fill two separate bowls with scoops of food. Specifically, the robot is asked to place an exact number of scoops of two different ingredients into different bowls. The robot needs to keep track of how many scoops have already been obtained per ingredient. This counting task has appeared in prior work (Torne et al., 2025), we modify it to require much longer-horizon reasoning by increasing the potential number of scoops and ingredients to scoop from. This task is challenging because the keyframes corresponding to each ingredient are nearly indistinguishable—piles look almost identical after each scoop—so missing or duplicating a keyframe can cause the high-level policy to misjudge its progress. We train and test with peanuts and jelly beans. *Evaluation metric.* Task completion is measured by the absolute value of the difference between the number of scoops requested and obtained, for each ingredient. Here, a lower metric is better. We also report the 0-1 success rate for satisfying the instruction.

**Dust & Replace.** In this task, the robot is asked to remove objects from a two-tiered shelf, pick up a duster, dust each shelf, and replace the objects to their original positions. Between dusting the two shelves, we return the duster to its reset position, making it unclear from recent context which shelf has already been dusted. This task is challenging because the robot must simultaneously remember

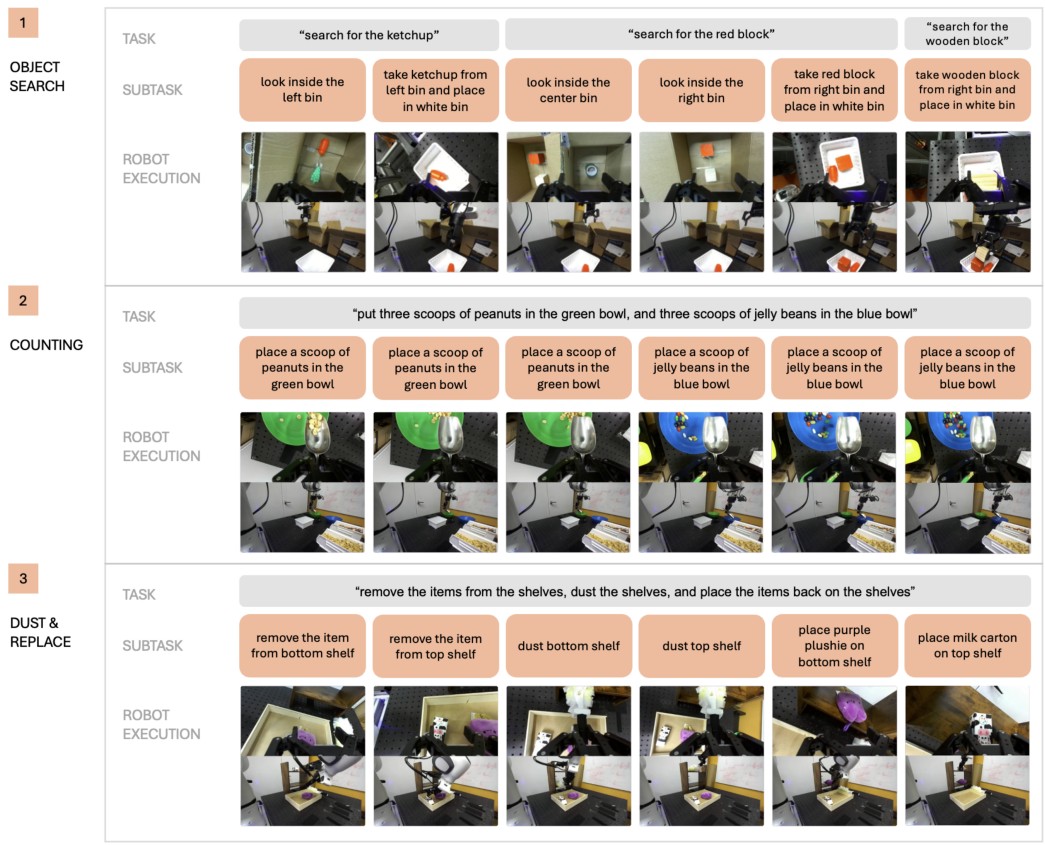

Figure 3: **Task used in our evaluation.** Across three domains, we evaluate complex instructions, intermediate subtasks, and keyframe predictions. We report performance across 20 trials per task per method.

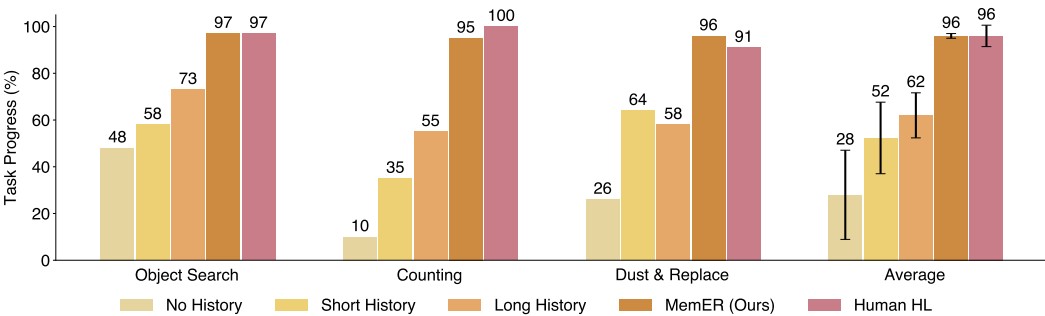

Figure 4: **Main Results.** Our method clearly outperforms the no history, short history (8 frames of context), and long history (32 frames of context) baselines on the three long-horizon memory-based tasks. It is on par with the human high level policy.

two types of information: the original locations of the objects and which shelf, if any, has already been dusted. We train and test with a set of 9 objects, which are various plushies. *Evaluation metric.* Task completion is measured by the binary success of each object being replaced correctly on the shelf and each shelf being dusted, for a max score of 4.

**Evaluation Setup:** Our robot setup resembles that within DROID (Khazatsky et al., 2025) having a Franka arm, parallel jaw gripper and two cameras: a third-person ZED camera and a wrist-mounted miniZED camera. For all tasks the $\pi_h$ operates at ~1Hz and the $\pi_l$ operates at ~2Hz. $\pi_l$ outputs an

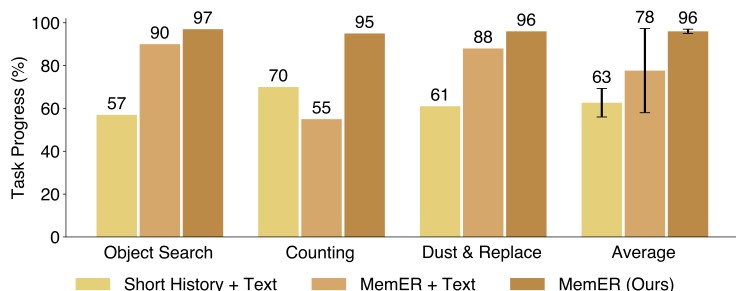

Figure 5: **Modality Results.** Using only images to represent the memory performs better than the baselines that use only text or text and images. We hypothesize that the high-level policy over-indexes on the text tokens in the memory, causing it to miss important details in the visual input.

| Method | Object Search | | Counting | Dust & Replace | | | |
|---|---|---|---|---|---|---|---|
| | # times object retrieved (↑) | # times used optimal path (↑) | # wrong scoops (↓) | Dust bottom shelf (↑) | Dust top shelf (↑) | Replace bottom object (↑) | Replace top object (↑) |
| MemER (Ours) | **59** | **57** | **1** | **20** | **19** | **18** | **20** |
| No history | 32 | 25 | 61 | 5 | 4 | 5 | 7 |
| Short History | 38 | 31 | 26 | 14 | 14 | 11 | 12 |
| Long History | 47 | 41 | 12 | 11 | 11 | 12 | 12 |
| Human HL | 58 | 58 | 0 | 19 | 19 | 18 | 17 |

Table 1: **Detailed Main Results.** Online evaluation of our method and the baselines for Q1. We provide task-specific evaluation metrics and the raw counts across 20 trials for each component of the task. Bold marks the best non-oracle method in each row. ↑ and ↓ indicate higher and lower is better, respectively.

action chunk $A_t$ of 15 actions, and we execute 8 open loop before replanning. The cameras stream $320 \times 180$ resolution images at 15Hz, but we subsample to 2Hz to input to hierarchical policy.

### 4.1 MAIN RESULTS

**Q1: To what extent can our approach tackle tasks that require memory compared to other methods?** All evaluated methods incorporate a $\pi_h$ and $\pi_l$, and the baselines change the input context of the high-level policy $\pi_h$ while using the same $\pi_l$. We compare to the following baselines: 1) No history: a memory-less high-level policy that only views the current frame (i.e. current robot foundation models), similar to (Shi et al., 2025) 2) Short History: a policy that views only the recent $N$ frames ($N = 8$ for our setup) 3) Long History: a policy that naively receives a longer context ($4\times$ that of Short History or $N = 32$ recent frames), and 4) Human HL: a human provides the correct subtasks. The Human HL policy establishes a rough estimate of the upper bound performance for all tasks, with failures stemming from the low-level policy. From Figure 4, we see that No History and Short History baselines perform poorly as all of the tasks simply require more context than what was provided. The Long History baseline shows that increasing the context can slightly help, but 32 frames ($\sim$16 seconds of memory) incurs an inference cost of 1 second, which approaches the limit of what can be tolerated in closed-loop settings. The Long History policy still performs on average 34% worse than MemER, necessitating strategies such as our method that consolidate keyframes rather than naively extending the context. Lastly, our method has $> 95\%$ on all tasks with the most common failure case being failures in the low-level policy executing the subtask, which can be rectified with better low-level correction data.

**Q2: How does our high-level policy, fine-tuned from an open-source VLM, compare to proprietary off-the-shelf VLMs?** Since our approach outperforms other selections of $\pi_h$ and performs similar to the Human HL policy, we investigate if our method is necessary given existing state-of-

| Method | Object Search | | Counting | | Dust & Replace | |
|---|---|---|---|---|---|---|
| | Trajectory acc. (↑) | Boundary acc. (↑) | Trajectory acc. (↑) | Boundary acc. (↑) | Trajectory acc. (↑) | Boundary acc. (↑) |
| MemER (Ours) | **0.80** | **0.76** | **0.67** | **0.65** | **0.87** | **0.86** |
| GPT–5 | 0.15 | 0.16 | 0.43 | 0.47 | 0.67 | 0.63 |
| Gemini Robotics–ER 1.5 | 0.21 | 0.23 | 0.13 | 0.14 | 0.19 | 0.22 |

Table 2: **Comparison with API-Based VLMs.** Offline evaluations of the per-task trajectory and boundary accuracy of subtask predictions between MemER, GPT-5, and Gemini Robotics-ER 1.5 (Team et al., 2025), to compare our finetuned high-level policy from an open-source VLM against proprietary VLMs.

| Method | Input Components | | | Object Search | | Counting | Dust & Replace | | | |
|---|---|---|---|---|---|---|---|---|---|---|
| | Short History | Image Keyframes | Text Subtasks | # times object retrieved (↑) | # times used optimal path (↑) | # wrong scoops (↓) | Dust bottom shelf (↑) | Dust top shelf (↑) | Replace bottom object (↑) | Replace top object (↑) |
| MemER (Ours) | ✓ | ✓ | ✗ | **59** | **57** | **1** | **20** | **19** | **18** | **20** |
| Short History + Text | ✓ | ✗ | ✓ | 40 | 28 | 10 | 16 | 16 | 7 | 10 |
| MemER + Text | ✓ | ✓ | ✓ | **59** | 49 | 13 | **20** | 18 | 17 | **20** |

Table 3: **Detailed Modality Results.** Online evaluation across methods ablating the textual modality. Bold marks the best method. ↑ and ↓ indicate higher and lower is better, respectively.

the-art VLMs may already have this capability. We test both GPT-5 and Gemini Robotics–ER 1.5 (Team et al., 2025) given the former's strong multimodal reasoning performance and the latter's robotics-specific agentic capabilities. Because the API latency for both ranged from 10-15 seconds, these API-based high-level policies led to complete failures when we deployed it in the same closed-loop evaluation as the other baselines, which require latencies of less than 1 second to react accordingly to the environment.

To still offer a means of comparison between $\pi_h$ and the API-based high-level policies, we designed an offline experiment using a held-out set of trajectories generated by the low-level policy commanded by ground-truth subtasks $l'_t$. This simulates closed-loop execution under realistic behaviors (i.e. retries after missed grasps, pauses, and jerky motions), while allowing the model to build its visual memory in the same way. We carefully craft the prompt to include specific task-relevant instructions and an explicit list of all possible actions that the low-level policy can follow, and ask the model to choose among them (Appendix C). Just like our setup, the model takes in the $N = 8$ most recent frames of context and selected keyframes $K_t$ at every timestep, and outputs the subtask $l'_t$ for the low-level policy to execute and candidate keyframes $J_t$. We measure *trajectory accuracy*, which is how often the correct subtask is predicted at each timestep in the trajectory, since we know the ground-truth subtask command that the low-level policy is executing at that moment. We also measure *boundary accuracy*, computed as the trajectory accuracy within a fixed window centered on transition points between subtasks. These are critical moments that expose the high-level policy's grasp of task progress by knowing when to move on to the next subtask; correct timing in transitioning between subtasks plays a major role in proper coordination with the low-level policy during deployment. From Table 2, we observe that both zero-shot API-based models perform poorly compared to our finetuned Qwen2.5-VL model, primarily failing by predicting too many non-informative candidate keyframes, reflecting its limited ability to identify which frames are truly useful. Consequently, even with a significantly stronger base VLM such as GPT-5 or Gemini Robotics–ER 1.5, the model lacks the capacity to interpret robot-specific perceptual cues and identify meaningful keyframes, resulting in less accurate subtask predictions and necessitating additional fine-tuning.

**Q3: How does representing memory via images compare to other modalities?**

We now discuss which modalities are best suited for building memory—visual, textual, or both. Storing memory in text offers natural benefits as it's interpretable and much more condensed. We test two additional methods that use text memory, in the form of the predicted subtask $l'_t$ that is associated with each of the selected keyframes in $K_t$: 1) Short History + Text uses the most recent

$N = 8$ frames and predicted subtasks and 2) MemER + Text interleaves the predicted subtasks and visual keyframes in memory. Table 3 shows the input for each baseline.

We see from Figure 5 (left) that relying on textual memory underperforms compared to our vision-only approach. Specifically, replacing the visual memory with text (Short History + Text) leads to the most significant performance drop. Furthermore, adding text to our visual memory (MemER + Text) provides no benefits, consistently under-performing across all tasks, especially the Counting task. We find that both the baselines' subtask predictions are brittle, largely due to overreliance on the most recently predicted subtask stored in memory. This leads to failures when policy retries or freezes shift the recent context out of distribution. In such cases, the model tends to overfit to the canonical ordering of subtasks observed in expert demonstrations and misidentifies the subtask being executed given the current environmental state. In contrast, directly grounding predictions in the current observation combined with the robust visual memory proves more reliable.

For the Short History + Text baseline, the language-based subtasks do not capture all of the information required to successfully complete the task. For example, in the Object Search task, the predicted language subtasks only specify the objects the robot has previously been asked to locate or is currently searching for, but have no reference to objects it has seen that may need to be retrieved in subsequent episodes. For the MemER + Text baseline, the model disproportionately attends to the text stored in memory, which can be incorrect for the reasons stated above, and subsequently ignores important information stored in visual memory. Such behavior has been noted before in (Zheng et al., 2025; il Lee et al., 2025). Thus, from our tasks, we find that visual memory alone provides the most robust representation, though exploring multimodal memory remains an interesting future direction.

## 5 Discussion and Future Work

We introduced MemER, a hierarchical vision–language–action framework that *scales memory via experience retrieval*. A high-level memory policy processes streamed observations, nominates keyframes to retain, and emits language subtasks that a low-level controller executes. A simple online consolidation strategy converts per-timestep candidate keyframes into a compact, stable episodic memory that is fed back into the high-level policy. Across three real-world, long-horizon manipulation domains, MemER significantly improves performance on tasks requiring minutes of recall while retaining low-latency inference and strong compatibility with existing VLA backbones.

Despite its benefits, our approach has several limitations. We continuously accumulate informative keyframes but currently lack a mechanism to discard them when they become too numerous—an issue that may arise for tasks requiring hours of memory. Enabling the high-level policy to reason about which keyframes to not only *add* but also *delete* for modifiable long-term memory is an exciting direction for future work. Aghajohari et al. (2025) proposes an approach that uses reinforcement learning to train an LLM to maintain a fixed-size memory state throughout its chain-of-thought reasoning. Adapting this idea could provide a promising way to endow MemER with a learned memory management system that includes an explicit *forgetting* mechanism. In addition, our memory is limited to visual observations; incorporating other sensory modalities such as tactile or audio is a promising extension. Finally, we study a single robot embodiment, and extending to mobile manipulation and multi-room tasks, where memory must interleave spatial mapping with episodic recall, would bring the system closer to human-like memory. We view MemER as a step toward robot policies that *decide what to remember* and *leverage those memories when needed* for effective long-horizon control.

## 6 Acknowledgments

This research was supported by ONR grant N00014-22-1-2621, the Robotics and AI Institute, and compute from the Stanford Institute for Human-Centered AI (HAI). We thank Dorsa Sadigh, John Yao, Moo Jin Kim, Lucy Xiaoyang Shi, Brian Kim, Homer Walke, Marcel Torne, Yuejiang Liu, Anikait Singh, Andy Tang, and Jennifer Grannen for their informative discussions. We thank Yuejiang Liu, John Yao, Alex Swerdlow, and Ria Doshi for their feedback on earlier versions of the paper. We also thank Physical Intelligence for providing beta access to the $\pi_{0.5}$ model. Ajay Sridhar is supported by the NSF Graduate Research Fellowship.

## 7 REPRODUCIBILITY STATEMENT

We link the code for creating the training data and practical real-world deployment on our website.

## 8 ETHICS STATEMENT

We are not presently aware of significant ethical issues arising from this work.

## 9 THE USE OF LARGE LANGUAGE MODELS (LLMs)

We only used LLMs to rephrase and polish the text for clarity and readability.

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

## A  MODEL INITIALIZATION AND HYPERPARAMETERS

**Training the High-Level Policy** The hyperparameters for fine-tuning the Qwen2.5-VL-7B-Instruct high-level policy are detailed in Table 4.

Table 4: Hyperparameters for High-Level Policy (Qwen2.5-VL-7B-Instruct) fine-tuning.

| Hyperparameter | Value |
|---|---|
| Learning Rate | 6e-5 |
| Optimizer | AdamW |
| $\beta_1$ | 0.9 |
| $\beta_2$ | 0.999 |
| Weight Decay | 0 |
| Gradient Clip Norm | 1.0 |
| LR Schedule | Cosine |
| Warmup Ratio | 0.05 |
| Batch Size | 256 |
| Training | 4500 gradient steps |
| Compute | 96 H200 GPU hours |
| Frozen Layers | Vision Encoder, Projection Layer |
| Trainable Layers | LLM Backbone |

**Training the Low-Level Policy** The hyperparameters for fine-tuning the $\pi_{0.5}$ low-level policy are detailed in Table 5. The model is fine-tuned from the public $\pi_{0.5}$ checkpoint trained on the DROID dataset (Khazatsky et al., 2025).

Table 5: Hyperparameters for Low-Level Policy ($\pi_{0.5}$) fine-tuning.

| Hyperparameter | Value |
|---|---|
| Learning Rate | 2.5e-5 |
| Optimizer | AdamW |
| $\beta_1$ | 0.9 |
| $\beta_2$ | 0.95 |
| Weight Decay | 0 |
| Gradient Clip Norm | 1.0 |
| LR Schedule | Cosine |
| Warmup Steps | 1000 |
| Batch Size | 128 |
| Training Steps | 18000 |
| Compute | 48 H200 GPU hours |

## B  DATA COLLECTION AND LABELING THE SUBTASKS.

For collecting the robot trajectory data we follow the data collection procedure with the Oculus teleoperation set in Khazatsky et al. (2025). To make the primitive labeling process for data collection as easy as possible, we generate the subtasks associated with the task before collection data. For instance, for the counting task, if we wanted to scoop 3 scoops of peanuts in the blue bowl and 2 scoop of jelly beans in the blue bowl, we would generated a list of subtasks for the whole trajectory. This includes a pick up scooper primitive, a primitive for each individual scoop, a reset scooper primitive between the scoops, and a drop scooper primitive. While collecting the data, we just need to follow what the current primitive is asking, and we only need to indicate when a primitive ends with a simple keyboard input. We also automate the randomization of the high-level task to avoid human biases when collecting data.

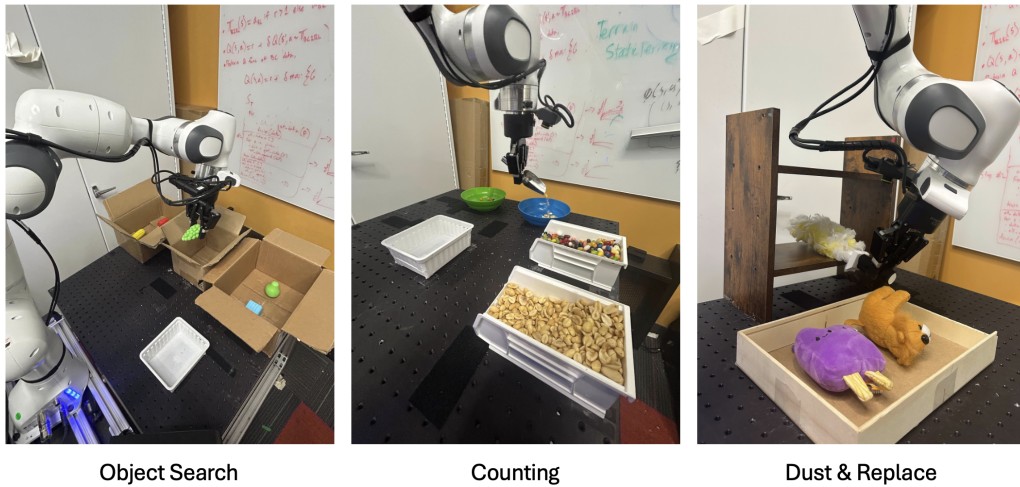

| Object Search | Counting | Dust & Replace |

Figure 6: Images of the three tasks.

## C PROMPTS FOR GPT-5 / GEMINI ROBOTICS–ER 1.5 EVALUATION

---

**Object Search System Prompt**

You are an AI assistant controlling a single-arm robot to search for specific objects amongst 3 bins. When exploring the bins for objects, look in the order of left bin, center bin, then right bin. You will receive images from two cameras: one for a global view and one on the robot's wrist for a detailed view. You will be provided with recent images that show the most recent actions the robot has executed. You will also be provided with selected keyframe images which are frames of particular importance from all the actions the robot has executed so far. Based on these, choose an action from the provided list for the robot to execute to best achieve the user's task instruction. Provide the exact action from the list without any explanation.

You will select your action from the following list:
- `look inside the <LOCATION> bin`
- `take the <OBJECT> from the <LOCATION> bin and place it in the white bin`

`<LOCATION>` is one of "left", "center", or "right".
`<OBJECT>` is one of "green tape", "red block", "corn", "baguette", "blue block", "fried chicken", "milk carton", "ketchup", "eraser", "grapes", "strawberry", "tomato", "pear", "wooden block", or "olive oil".

You will also return a list of values from 1-8 to index which of the frames from the most recent actions seem to be of particular importance for the robot to remember. For this task, recalling what objects are in each bin is critical, so you should return a list of indices, if any, from the most recent frames that provides a good view of a bin.

Return a JSON with:
- `current_subtask`: the action that should be executed at the current timestep, selected from the above list using the stated `<OBJECT>` and `<LOCATION>` values
- `keyframe_positions`: list of frame positions from 1-8, if any, from the recent frames to keep track of which objects are in each bin

---

**Counting System Prompt**

You are an AI assistant guiding a single-arm robot to obtain a specific amount of scoops of two different ingredients. You will reset the scooper between each scoop, and drop the scooper when all scoops across both ingredients have been obtained. You will receive images from two cameras: one for a global view and one on the robot's wrist for a detailed view. You will be provided with recent images that show the most recent actions the robot has executed. You will also be provided with selected keyframe images which are frames of particular importance from all the actions the robot has executed so far. Based on these, choose an action from the provided list for the robot to execute to best achieve the user's task instruction. Provide the exact action from the list without any explanation.

You will select your action from the following list:
- `pick up the scooper`
- `place a scoop of <OBJECT> in the <COLOR> bowl`
- `reset scooper position`
- `drop the scooper`

`<OBJECT>` is one of "peanuts" or "jelly beans".
`<COLOR>` is one of "green" or "blue".

You will also return a list of values from 1-8 to index which of the frames from the most recent actions seem to be of particular importance for the robot to remember. For this task, recalling how many scoops of each ingredient have been obtained is critical, so you should return a list of indices, if any, from the most recent frames that provides a good view of a completed scoop.

Return a JSON with:
- `current_subtask`: the action that should be executed at the current timestep, selected from the above list using the stated `<OBJECT>` and `<COLOR>` values
- `keyframe_positions`: list of frame positions from 1-8, if any, from the recent frames to keep track of scoops

**Dusting System Prompt**

You are an AI assistant guiding a single-arm robot to take an object off each shelf (bottom shelf then top shelf), pick up the duster, dust the bottom shelf, reset the duster, dust the top shelf, put down the duster, and replace the objects back to their original places (bottom shelf then top shelf). You will receive images from two cameras: one for a global view and one on the robot's wrist for a detailed view. You will be provided with recent images that show the most recent actions the robot has executed. You will also be provided with selected keyframe images which are frames of particular importance from all the actions the robot has executed so far. Based on these, choose an action from the provided list for the robot to execute to best achieve the user's task instruction. Provide the exact action from the list without any explanation.

You will select your action from the following list:
- `remove the object on the bottom shelf`
- `remove the object on the top shelf`
- `pick up duster`
- `dust bottom shelf`
- `reset duster`
- `dust top shelf`
- `put down duster`
- `place the <OBJECT> on the bottom shelf`
- `place the <OBJECT> on the top shelf`

`<OBJECT>` is one of "panda plushie", "purple plushie", "zebra plushie", "elephant plushie", "lion plushie", "smily face ball", "hello kitty plushie", "baby shoe", "milk carton".

You will also return a list of values from 1-8 to index which of the frames from the most recent actions seem to be of particular importance for the robot to remember. For this task, recalling where the items were originally placed on the shelves and which shelves have been dusted is critical, so you should return a list of indices, if any, from the most recent frames that provides a good indication of either.

Return a JSON with:
- `current_subtask`: the action that should be executed at the current timestep, selected from the above list using the stated `<OBJECT>` values
- `keyframe_positions`: list of frame positions from 1-8, if any, from the recent frames to keep track of where the objects were originally placed on the shelves and which shelves have been dusted

## D    INFERENCE SPEED AND MEMORY COST

An important aspect of using keyframes as a sparse memory representation is the ability to maintain fast inference and low memory usage for solving tasks that require reasoning over hundreds of frames. We run all of our experiments on an NVIDIA GeForce RTX 4090 GPU with our finetuned Qwen2.5-VL-Instruct-7B model. In Figure 7, we show how the latency for the high-level policy changes as we increase the number of keyframes in its context $|K_t|$ from 0 to 8 (we also keep the recent context at $|R_t| = 8$). We see that the inference speed always stays below 0.8s per prediction. Empirically, we found any high-level policy that can predict at 1Hz or faster to perform the best on real-world deployments, so we are well within this limit. Additionally, our VRAM usage is within the 24GB limit for a 4090, so we can run our high-level policy on a single card.

However, if we were to use a naive method of retaining long-range dependencies by simply keeping track of more recent frames, we see in Figure 8 that the inference cost quickly blows up. Specifically, after increasing the number of recent frames beyond 32, the inference cost exceeds 1.0s (the 1Hz threshold), so it would not coordinate well with the low-level controller in a real-world setting.

In Table 6, we show the inference speed for the low-level policy, $\pi_{0.5}$. $\pi_{0.5}$ can also run on a 4090 GPU well above the desired speed of 2Hz. We only need **two 4090s** to run our hierarchical policy.

| Model Configuration | Inference Time (s) | VRAM (GB) |
|---|---|---|
| $\pi_{0.5}$ | $0.088 \pm 0.001$ | 6.25 |
| MemER (8 recent + 8 keyframes) | $0.787 \pm 0.066$ | 15.93 |
| No History (1 recent frame) | $0.532 \pm 0.065$ | 15.55 |
| Short History (8 recent frames) | $0.591 \pm 0.064$ | 15.64 |
| Long History (32 recent frames) | $0.874 \pm 0.065$ | 16.01 |

Table 6: Comparison of inference speed and VRAM usage across models for the high-level (Qwen2.5-VL-Instruct-7B) and low-level ($\pi_{0.5}$) policy on a 4090 GPU. We run 20 trials per value.

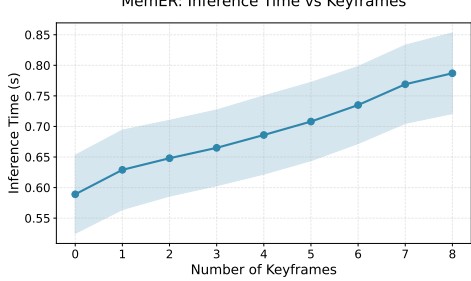
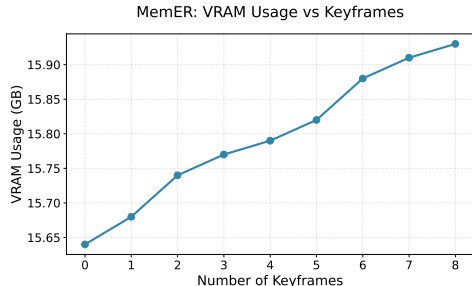

Figure 7: Plot of the average inference inference speed (**Left**) and VRAM usage (**Right**) for the MemER high-level policy. We evaluate Qwen2.5-VL-7B-Instruct for the high-level policy. All configurations include 8 recent frames ($|R_t| = 8$); the x-axis shows the number of *additional* keyframes added to context. We run 20 trials for each data point.

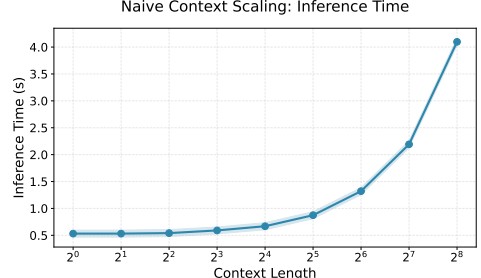 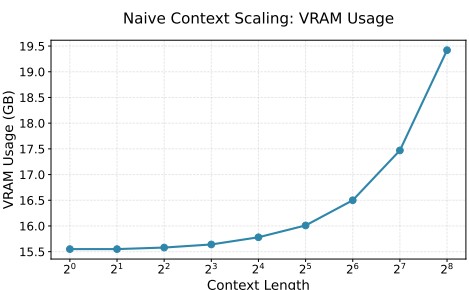

Figure 8: Plot of the average inference speed **(Left)** and VRAM usage **(Right)** for naively scaling context in the high-level policy. We evaluate Qwen2.5-VL-7B-Instruct for the high-level policy. This baseline uses only recent frames (no additional keyframes); the x-axis shows the total context length $|R_t|$ from 1 ($2^0$) to 256 ($2^8$). We run 20 trials for each data point.

## E  ANNOTATION RULES FOR KEYFRAMES

As described in Section 3.3, we use a simple annotation rule for each subtask to build the set of keyframes that constitute the ground-truth targets. We take the last frame of the following subtasks:

- **Object Search**
  - `"look inside the <LOCATION> bin"`
- **Counting**
  - `"place a scoop of <OBJECT> in the <COLOR> bowl"`
- **Dust & Replace**
  - `"remove the object on the bottom shelf"`
  - `"remove the object on the top shelf"`
  - `"dust bottom shelf"`
  - `"dust top shelf"`
  - `"place the <OBJECT> on the bottom shelf"`
  - `"place the <OBJECT> on the top shelf"`

The last frames of these subtasks represent what the policy needs to remember.

## F  CROSS-TASK OBJECT GENERALIZATION

To evaluate the benefits of multi-task training, we compare a single-task version of MemER (separate policy trained for each task) with our multi-task version. For these experiments, we finetune Qwen2.5-VL-Instruct-7B for the single- and multi-task versions of the high-level policies. We evaluate on our object-centric tasks: the Object Search and Dust & Replace tasks, using objects from Figure 10. First, we establish a baseline by evaluating both versions of MemER on their original task setups. As shown in Figure 9 (left), their performance is roughly similar.

The primary benefit of multi-task training is revealed when evaluating cross-task object generalization. For this evaluation, we swap all of the objects between the tasks (e.g., using Object Search objects for the Dust & Replace task, and vice-versa). This creates new object-task combinations that the models have not seen during training. The results in Figure 9 (right) demonstrate the clear advantage of using the multi-task model (82% success), generalizing much more effectively to out-of-domain combinations than the single-task version (59% success). This demonstrates that our method learns a generalizable skill of "what to remember" that transfers to new scenarios, rather than just overfitting to the original training demonstrations.

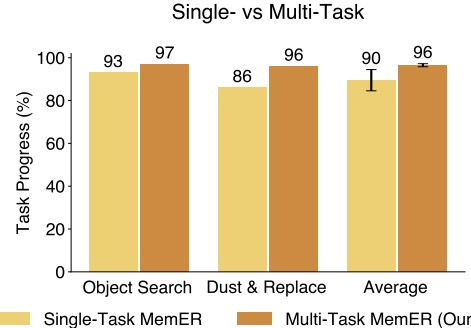
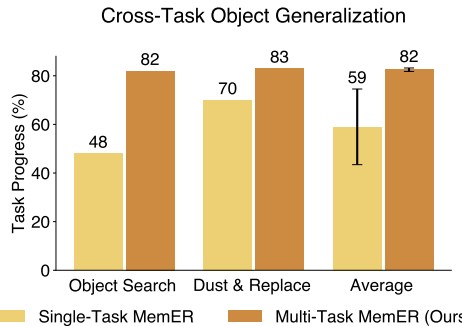

Figure 9: **(Left) Single- vs Multi-Task Results.** The performance of the single- and multi-task versions of MemER on the Object Search and Dust & Replace task are similar. **(Right) Cross-Task Object Generalization Results.** We swap objects between the Object Search and Dust & Replace (see Figure 10) tasks during evaluation. The multi-task policy can generalize to the new object-task combinations during evaluation despite never seeing them in training.

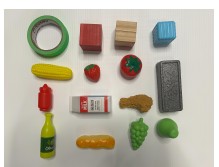
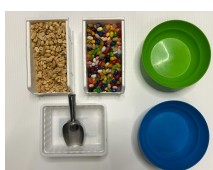
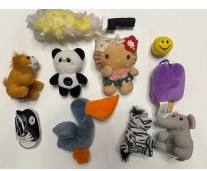

Figure 10: **(Left)** Objects for the Object Search task. **(Center)** Objects for the Counting task. **(Right)** Objects for the Dust & Replace task.

# G  KEYFRAME SELECTION ALGORITHM

---

**Algorithm 1** Selecting Keyframes from Candidates

---

**Input:**  A sequence of candidate keyframe sets $\boldsymbol{J'_{0:t}} = (\boldsymbol{J_0}, \boldsymbol{J_1}, \ldots, \boldsymbol{J_t})$
        The merge distance $d$
**Output:**  A list of the selected keyframes $\boldsymbol{K_t}$

 1: **function** BUILDVISUALMEMORY($\boldsymbol{J'_{0:t}}$, $d$)
 2:    $\boldsymbol{G_{0:t}} \leftarrow$ Sort(GetIndicesFromFrames($\boldsymbol{J'_{0:t}}$))            ▷ Extract the temporal indices.
 3:    **if** $\boldsymbol{G_{0:t}}$ is empty **then**               ▷ Handle case with no candidates.
 4:        **return** $\emptyset$
 5:    **end if**
 6:    $Clusters \leftarrow []$
 7:    $C_{current} \leftarrow [\boldsymbol{G_{0:t}}[0]]$
 8:    **for** $i = 1$ to $|\boldsymbol{G_{0:t}}| - 1$ **do**            ▷ Build the clusters.
 9:        **if** $\boldsymbol{G_{0:t}}[i] - \boldsymbol{G_{0:t}}[i-1] \leq d$ **then**
10:            Append $\boldsymbol{G_{0:t}}[i]$ to $C_{current}$
11:        **else**
12:            Append $C_{current}$ to $Clusters$
13:            $C_{current} \leftarrow [\boldsymbol{G_{0:t}}[i]]$         ▷ Start a new cluster.
14:        **end if**
15:    **end for**
16:    Append $C_{current}$ to $Clusters$
17:    $T_{selected} \leftarrow []$
18:    **for** each cluster $C$ in $Clusters$ **do**     ▷ Select the median index of each cluster.
19:        $i_{median} \leftarrow$ Median($C$)
20:        Append $i_{median}$ to $T_{selected}$
21:    **end for**
22:    $\boldsymbol{K_t} \leftarrow$ GetFramesFromIndices($T_{selected}$)
23:    **return** $\boldsymbol{K_t}$
24: **end function**

---

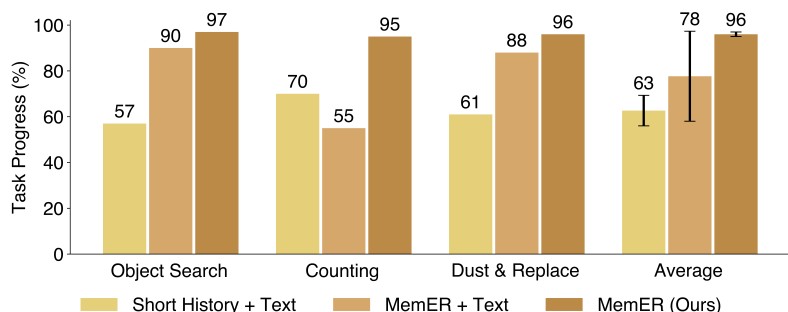

Figure 11: Merging our fine-tuned high-level policy's weights with its base model's weights improves or maintains performance on all tasks.

## H  HIGH-LEVEL POLICY PARAMETER MERGING

An important factor contributing to the success of our policy is the strong video understanding prior in Qwen2.5-VL-7B-Instruct (Bai et al., 2025). However, training the high-level policy to accurately predict the language subtasks used by the low-level policy requires roughly 5,000 gradient steps. After this amount of fine-tuning, the high-level policy tends to lose some robustness to low-level policy freezes and retry behaviors, due to its training data consisting solely of optimal expert demonstrations. Concurrent work suggests that linearly interpolating the weights of a generalist pretrained model with those of the same model fine-tuned on narrow, task-specific data can help preserve the pretrained model's robustness and generalization, while still allowing adaptation to the new task (Anonymous, 2025). We find this also applies to the high-level policy. Specifically, we set the weights of our high-level policy as:

$$\theta = (1 - \alpha) \cdot \theta_{\text{pre}} + \alpha \cdot \theta_{\text{ft}} \tag{3}$$

where $\theta_{\text{pre}}$ is the weights of Qwen2.5-VL-7B-Instruct and $\theta_{\text{ft}}$ is the weights of this model fine-tuned on all three memory-based tasks. We follow Anonymous (2025) and set $\alpha = 0.8$ for all baselines we test. Figure 11 shows that model merging improves or maintains performance across all tasks.

## I  FREQUENCY DOMAIN-BASED CLUSTERING EXPERIMENTS

We originally explored more heuristic-based methods for selecting keyframes, but found them to be much less reliable than semantically selected keyframes. MemER's keyframes come from a trained high-level VLM that nominates task-relevant frames, and the clustering step consolidates these into a compact memory. On the other hand, frequency-domain methods operate on low-level intensity/spectral changes, so they mostly detect visual motion rather than subtask boundaries that correspond to meaningful visual states. Because the VLM sees language and multi-view context, it can learn to ignore viewpoint noise and only nominate genuinely informative frames. Frequency-domain changes spike on any large camera motion or background shift, so they over-trigger on irrelevant movement in egocentric manipulation.

To illustrate this, we ran UniDomain's (Ye et al., 2025) clustering method using energy-based extrema in a sliding window, and observed that the selected keyframes are much less informative. We take a random Counting Task demonstration as an example and include the results below. To construct a memory buffer that is the same size as our method, the window size needed to be >350, which means that a lot of salient information will be lost if it occurs within 350 frames of a local minimum/maximum (Figure 12). If we allow the window to be smaller but still sparse enough such that the high-level policy runs at 1Hz (32 frames of memory, window size of 35), we get incredibly noisy and uninformative keyframes (Figure 13).

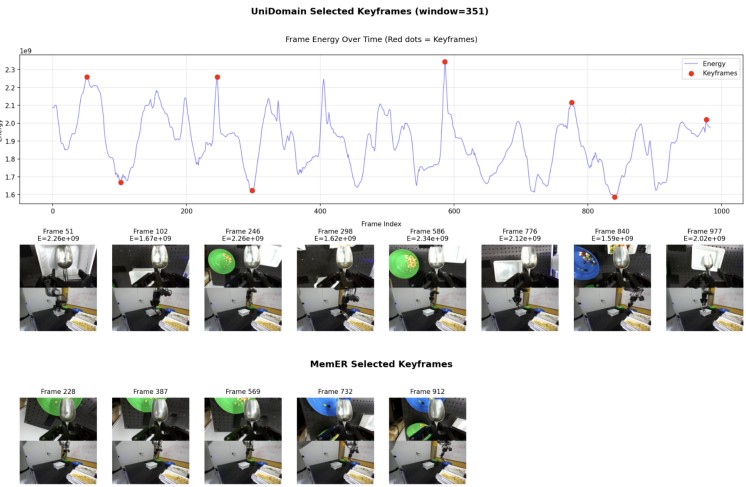

Figure 12: **(Top)** UniDomain's clustering method with a window size of 351. **(Bottom)** MemER's selected keyframes.

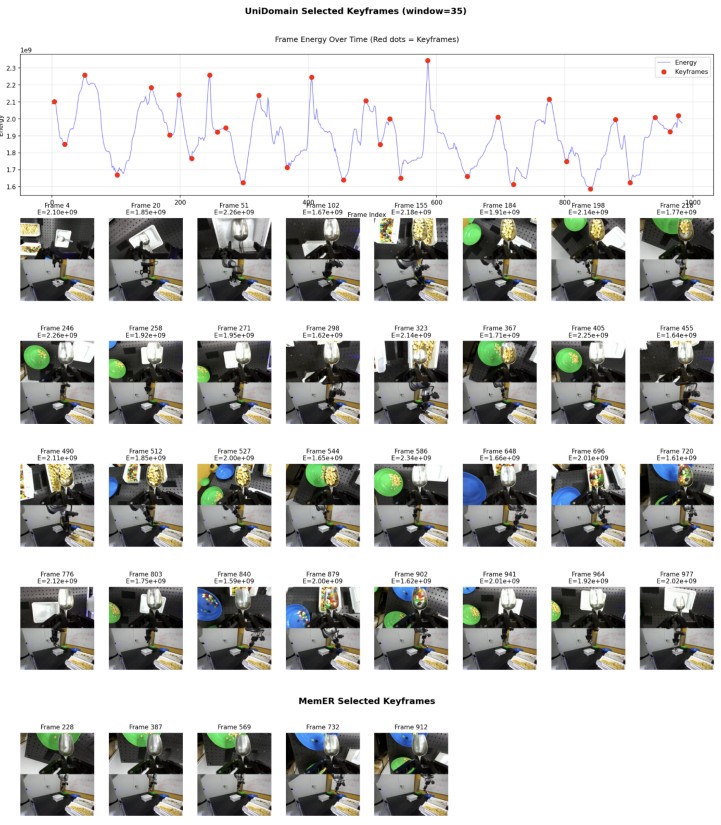

Figure 13: **(Top)** UniDomain's clustering method with a window size of 35. **(Bottom)** MemER's selected keyframes.

## J    ATTEMPTS AT A UNIFIED HIERARCHICAL MODEL

Additionally, our approach trains two models for the hierarchical policy Shi et al. (2025), while a more elegant method would use the same set of weights for a unified policy Black et al. (2024). We initially tried to train a unified model by fine-tuning $\pi_{0.5}$ for subtask prediction from videos. This failed as $\pi_{0.5}$ lacked the proper video understanding pretraining to memory-based subtask prediction from just 50 demos. We also tried to train Qwen2.5VL to predict low-level actions in addition to subtasks and keyframes, but we found it difficult to balance all training objectives on our limited dataset. However, it would be worthwhile to take the ideas from MemER, such as keyframe prediction and management, and scale them to large pretraining runs for VLAs to train the next generation of memory-based VLAs.

We found that existing pre-trained models are quite specialized: $\pi_{0.5}$ has strong action priors but very poor video-understanding priors, while Qwen2.5-VL has the reverse behavior. A unified model struggled to effectively learn both memory-aware subtask prediction and action prediction from our limited data (50 demos/task). We attempted two unified variants:

**Fine-tuning $\pi_{0.5}$ (VLA) for high-level memory reasoning:** We tried fine-tuning $\pi_{0.5}$ to predict subtasks and keyframes from 8 frames of context (the setup for the high-level policy in MemER) in addition to action predictions. This failed, as the model lacked the necessary video-understanding pre-training to reason about long-horizon context.

**Fine-tuning Qwen-VL (VLM) for low-level actions**: We tried fine-tuning Qwen2.5-VL to predict low-level actions using the FAST tokenizer, in addition to subtasks and keyframes. We found it extremely unstable to train with both the action-generation and video-planning losses.

Our hierarchical design is a pragmatic solution that leverages the distinct strengths of both pre-trained models, and this modularity is what enables MemER to succeed in complex, multi-minute tasks from only 50 demonstrations. Moreover, our method is compatible with existing VLA models to allow the system to efficiently reason over long-horizon dependencies.

