# OpenReview forum: "Scaling up Memory for Robotic Control via Experience Retrieval"
_ICLR.cc/2026/Conference — ICLR 2026 Poster_

### Official Review · Reviewer_uYHS · 2025-10-28

**Soundness:** 3
**Presentation:** 3
**Contribution:** 3
**Rating:** 6
**Confidence:** 4

**Summary:**

This work addresses the problem that robotic policies often rely only on the current observation or a few recent frames and thus lack long-term memory. It proposes a hierarchical policy framework: the high-level policy selects keyframes or memories from past experiences and generates language-based subtasks or instructions based on both the current observation and the retrieved keyframes; the low-level policy then receives the current image, the robot’s current state, and the subtask produced by the high-level policy to execute the concrete actions.

**Strengths:**

1. The system design is simple, effective, and scalable.

2. The results show a significant improvement in long-horizon task success rates.

3. The writing is clear and easy to follow.

**Weaknesses:**

1. The evaluation of scalability aspects such as memory size and retrieval latency could be more detailed.

2. Providing a conceptual comparison among different types of approaches addressing the long-horizon problem would offer readers deeper insights.

**Questions:**

1. How does the high-level policy determine which frames are keyframes? When the task types or scenes vary significantly, is this keyframe selection mechanism still generalizable, or does it require task-specific tuning?

2. The paper includes baselines such as “Short History (8 frames)” and “Long History (32 frames).” I’m curious why the long-history setup leads to such a large improvement. Moreover, since simply adding more historical frames performs worse than MemER, why does “more history” not yield the same gains as the keyframe retrieval mechanism?

3. After the high-level module generates subtasks and keyframes, isn’t the way the low-level policy directly uses this information somewhat too simple?

4. Although the paper proposes using memory to tackle long-horizon problems, it does not compare MemER conceptually with other VLA-based or related long-horizon approaches [1, 2, 3]. Even if experimental comparison is difficult, a conceptual discussion would provide readers with deeper insights.

5. The paper mentions that keyframes are accumulated into memory but currently lacks a mechanism for “deletion” when the memory becomes too large. Are there any promising methods to address this issue?

---

> ### Author Response · Authors · 2025-11-19
> **Response to Reviewer uYHS (Part 1/2)**
>
> We thank Reviewer uYHS for their constructive feedback. We have revised the paper **(updates in red text)** and conducted new experiments that we believe address their concerns.
>
> ## Compute Analysis
>
> >The evaluation of scalability aspects such as memory size and retrieval latency could be more detailed.
>
> We have added new benchmark results to Appendix D detailing the inference speed (latency) and VRAM usage for our high-level policy when conditioned on 0 to 8 keyframes on a 4090 GPU. We use Qwen2.5VL-7B-Instruct as our new high-level policy. The subtask prediction is **~0.8s** when there are 8 keyframes + 8 recent frames in the context (16 frames total in context), which is within our 1Hz threshold for real-world deployment. The VRAM usage is around **16GB**, which is within the 24GB limit of a 4090, making it deployable on a single card.
>
> | Model Configuration | Inference Time (s) | VRAM (GB) |
> | :--- | :---: | :---: |
> | $\pi_{0.5}$ | $0.088 \pm 0.001$ | $6.25$ |
> | MemER (8 recent + 8 keyframes) | $0.787 \pm 0.066$ | $15.93$ |
> | No History (1 recent frame) | $0.532 \pm 0.065$ | $15.55$ |
> | Short History (8 recent frames) | $0.591 \pm 0.064$ | $15.64$ |
> | Long History (32 recent frames) | $0.874 \pm 0.065$ | $16.01$ |
>
> ## Conceptual Comparison
>
> > Providing a conceptual comparison among different types of approaches addressing the long-horizon problem would offer readers deeper insights.
>
> and
>
> > Although the paper proposes using memory to tackle long-horizon problems, it does not compare MemER conceptually with other VLA-based or related long-horizon approaches [1, 2, 3]. Even if experimental comparison is difficult, a conceptual discussion would provide readers with deeper insights.
>
> Thank you for this suggestion. We have expanded our Related Work (Sec 2) with a more detailed conceptual comparison to other long-horizon approaches in the “Foundation Models and Long-Horizon Tasks in Robotics” subsection.
>
> Most long-horizon VLAs use hierarchical structures via latent embeddings (Shentu et al., 2025; Wu et al., 2024), language (Shi et al., 2025), or waypoints (Li et al., 2025b). While enabling useful temporal abstraction for long-horizon tasks  (Zhang et al., 2023), they typically lack persistent memory. MemER adds a sparse, retrieval-based memory mechanism ($K_t$) to preserve long-range dependencies with fast inference, which is essential for multi-minute tasks where standard methods fail. We distinguish MemER from these long-horizon approaches: while they address temporal abstraction (planning), they generally assume Markovian states, missing the episodic recall (remembering) required for these tasks.
>
> Could you provide additional clarity on what you were referring to with [1,2,3] so we can offer a more detailed comparison with those approaches?
>
> ## Keyframe Selection
>
> > How does the high-level policy determine which frames are keyframes?
>
> This is a two-part process: (1) how we generate ground-truth labels for training, and (2) how the policy autonomously selects keyframes during deployment.
>
> 1. **How we generate labels for training:** We use a lightweight, **semi-automatic heuristic** that takes only **~5 minutes per task.** We determine a simple annotation rule per subtask - deciding whether or not to keep the last frame of that subtask segment as a ground-truth keyframe, since these transition points usually contain a visually informative state. For example:
>
>     * `look inside <LOCATION> bin` $\rightarrow$ Rule: Save the last frame (to remember what was inside).
>     * `remove object from...` $\rightarrow$ Rule: Save the last frame (to remember the object's original spot).
>     * `reset scooper` $\rightarrow$ Rule: Save no frame (this action is not for memory).
>
>     This simple ruleset is then automatically applied to all 50 demonstrations of each task. We have clarified this process in Sec 3.3 and added the full ruleset to Appendix E.
>
> 2. **How the policy selects during deployment:** The trained VLM is fully autonomous. At every timestep, it processes the most recent 8 frames and nominates candidate keyframes. These nominations are passed to our task-agnostic Keyframe Filter (detailed in Sec 3.2, Fig. 2, the videos under the "Task Visualizations" section on the [website](https://memer-policy.github.io/), and the newly added Algorithm 1 in Appendix G), which consolidates these nominations, filters for redundancy, and builds the compact memory $K_t$.
>
> **References**
>
> Shi et al. Hi robot: Open-ended instruction following with hierarchical vision-language-action models, 2025. URL https://arxiv.org/abs/2502.19417.
>
> Shentu et al. From llms to actions: Latent
> codes as bridges in hierarchical robot control, 2025. URL https://arxiv.org/abs/2405.
> 04798.
>
> Wu et al. Discrete policy: Learning disentangled action space for multi-task robotic manipulation. arXiv
> preprint arXiv:2409.18707, 2024.
>
> Li et al. Hamster: Hierarchical
> action models for open-world robot manipulation, 2025b. URL https://arxiv.org/abs/2502.
> 05485.

---

> > ### Comment · Reviewer_uYHS · 2025-11-21
> >
> > The evaluation of scalability aspects, such as memory size and retrieval latency, as well as keyframe generation, has already been addressed. The authors clearly conveyed this in the paper, and it is done very well.
> >
> > And I sincerely apologize for having overlooked the related references [1, 2, 3]. They are:
> >
> > [1] Plan-Seq-Learn: Language Model Guided RL for Solving Long-Horizon Robotics Tasks. ICLR 2024.
> > [2] SPIRE: Synergistic Planning, Imitation, and Reinforcement for Long-Horizon Manipulation. CoRL 2024.
> > [3] Long-VLA: Unleashing Long-Horizon Capability of Vision Language Action Model for Robot Manipulation. CoRL 2025.

---

> ### Author Response · Authors · 2025-11-19
> **Response to Reviewer uYHS (Part 2/2)**
>
> >When the task types or scenes vary significantly, is this keyframe selection mechanism still generalizable, or does it require task-specific tuning?
>
> We have run new experiments and replaced our single-task models with a single, **multi-task high-level policy** (Sec 3.3, Fig. 4). Our method still significantly outperforms all baselines on the three long-horizon memory-based tasks, achieving **>95%** progress on all tasks. We then evaluated the single- and multi-task models on a new cross-task generalization experiment (Fig. 9), testing its ability to handle novel task/object combinations (e.g., using Object Search objects in the Dust & Replace task). Our multi-task model outperforms the single-task models **(82% vs. 59% avg. success)** on these out-of-domain combinations. This illustrates our method's ability to learn a generalizable skill of "what to remember" that transfers to new scenarios, rather than overfitting to the original training demonstrations. The only task-specific part is the one-time setup of the keyframe rules, which takes ~5 minutes per task and incurs negligible overhead.
>
> > The paper includes baselines such as “Short History (8 frames)” and “Long History (32 frames).” I’m curious why the long-history setup leads to such a large improvement. Moreover, since simply adding more historical frames performs worse than MemER, why does “more history” not yield the same gains as the keyframe retrieval mechanism?
>
> Your analysis is exactly right. The difference is dense, sliding window vs. sparse, persistent memory. "Long History (32 frames)" is just a longer sliding window (~16 seconds). MemER is a selective, long-term memory that saves keyframes from the entire history, even minutes ago. Our Dust & Replace task (videos on [website](https://memer-policy.github.io/)) shows this perfectly:
> - "Long History (32f)" is better than “Short History (8f)” because 16s is long enough to remember recent actions (e.g., "just dusted the bottom shelf"), which the 8f policy forgets.
> - "Long History (32f)" still fails because the objects' original positions (from 60+ seconds ago) have dropped out of the 32-frame window.
> - MemER Succeeds because it saves both the original position keyframe from 2 mins ago and the "dusted shelf" keyframe from 30s ago.
>
> This is a key finding: naively extending the sliding window is not a scalable solution. It's computationally expensive and still fails at true long-range dependencies.
>
> > After the high-level module generates subtasks and keyframes, isn’t the way the low-level policy directly uses this information somewhat too simple?
>
> Our goal was to create a memory framework that can be modularly plugged into the large ecosystem of pre-trained VLAs [1, 2]. By using the standard language string interface, our memory-aware high-level policy can be adopted by any VLA conditioned on language. While the interface (a text string) is simple, the information it represents is a highly compressed semantic summary of the complex visual history processed by the high-level policy. While other modalities are a promising future direction [3, 4], our work proves a simple language interface is efficient and sufficient for these complex tasks.
>
> >The paper mentions that keyframes are accumulated into memory but currently lacks a mechanism for “deletion” when the memory becomes too large. Are there any promising methods to address this issue?
>
> This is an excellent point. Our current work focuses on tasks on the scale of minutes, where a full memory (up to 8 keyframes in practice) is not a bottleneck. You are right that for hour-long tasks, a deletion mechanism would be needed. This is an exciting direction that we have begun exploring, and there are two promising approaches: 1) train the VLM to identify which keyframes are "stale" and can be discarded given new updates in memory, 2) use reinforcement learning to train the VLM to manage a memory with a fixed-size budget [5]. We have added a discussion of these future directions to the Discussion section of our revised paper.
>
> Thank you again for your insightful review. We hope our changes have addressed your concerns, and we look forward to further discussion. Please let us know if you have any more questions or concerns.
>
>
>
> **References**
>
> [1] Physical Intelligence Team. π0.5: a vision-language-action model with open-world generalization, 2025. URL https://arxiv.org/abs/2504.16054.
>
> [2] NVIDIA et al. Gr00t n1: An open foundation model for generalist humanoid robots, 2025b. URL https://arxiv.org/abs/2503.14734.
>
> [3] Li et al. Hamster: Hierarchical action models for open-world robot manipulation, 2025c. URL https://arxiv.org/abs/2502.05485.
>
> [4] Zhang et al. Peek: Guiding and minimal image representations for zero-shot generalization of robot manipulation policies, 2025. URL https://arxiv.org/abs/2509.18282.
>
> [5] Aghajohari et al. The markovian thinker: Architecture-agnostic linear scaling of reasoning, 2025. URL https://arxiv.org/abs/2510.06557.

---

> > ### Comment · Reviewer_uYHS · 2025-11-21
> >
> > Thank you for the authors’ response. I believe the proposed method is a valuable attempt at enhancing hierarchical VLA with an **explicit** memory module. I also hope to see future developments toward more effective real-world keyframe detection and improved integration with lower-level VLAs.

---

> > > ### Author Response · Authors · 2025-11-24
> > >
> > > We thank Reviewer uYHS for the positive feedback and for confirming that our scalability experiments and updated evaluation address their concerns, as well as recognizing our method as a valuable step toward enhancing hierarchical VLAs with explicit memory. We have also added their specific references to our updated Related Work section. To elaborate further:
> > > - **Plan-Seq-Learn** [1] uses an LLM planner, a motion planner, and RL to learn low-level skills from scratch in simulation, so it tackles long horizons primarily via symbolic task decomposition without any learned memory over extended histories.
> > > - **SPIRE** [2] similarly combines task and motion planning to structure long-horizon manipulation into segments, which are first trained by imitation and then refined with RL, again focusing on learning local skills within a planner-defined hierarchy rather than on remembering any past visual context.
> > > - **Long-VLA** [3] uses an end-to-end VLA with a phase-aware masking module to better chain skills within a fixed visual context window, but it still operates within that bounded context and does not maintain an external episodic memory over minutes of interaction.
> > >
> > > In contrast, MemER introduces a high-level policy that **learns which frames to remember over entire multi-minute episodes** for memory-aware subtask prediction, endowing VLAs with persistent visual memory.
> > >
> > > Thank you for these suggested references - we hope this conceptual comparison resolves your remaining questions and strengthens your assessment of the paper.

---

> > > > ### Comment · Reviewer_uYHS · 2025-11-25
> > > >
> > > > Thank you for your response. I hope that in the final version, the authors can also compare with current long-horizon approaches to provide readers with more insights.

---

### Official Review · Reviewer_JibS · 2025-10-30

**Soundness:** 3
**Presentation:** 3
**Contribution:** 2
**Rating:** 4
**Confidence:** 2

**Summary:**

This paper presents MemER, a hierarchical architecture for Vision-Language-Action (VLA) modeling. The low-level policy is a general VLA model, while the high-level policy employs a Vision-Language Model (VLM) to predict primitives and identify key frames. The core contribution is the proposed experience retrieval mechanism, which operates at the high-level policy, enabling the VLM to leverage critical historical information for more accurate primitive prediction.

**Strengths:**

- The paper is easy to follow, and the figures and tables are clear and easy to understand.
- There is a strong motivation for this work. Memory is essential for real-world, long-horizon tasks, a need that is often overlooked in existing VLA literature.
- The approach is simple and effective. The low-level VLA model requires minimal or even no training, with the bulk of the effort focused on fine-tuning the high-level VLM. This results in a low overall training cost.
- The experimental validation provided is thorough and comprehensive.

**Weaknesses:**

- The architecture of MemER appears overly simple. The core advancements appear to lie primarily in the training methodology and data preparation for the high-level VLM, and these specific improvements do not seem to be particularly novel. This raises concerns about the overall innovation of the paper.

- The data preparation phase seems to be highly resource-intensive, relying on manual annotation of primitives and the segmentation of trajectories.

**Questions:**

- Could the authors more clearly articulate the specific innovative contributions of this paper? Given that the architecture is simple and the high-level training methods appear familiar, a clearer distinction from existing work is necessary to establish the technical merit.

- The authors discuss several related VLA models that utilize memory in the Related Works section. Why are these models not included as baselines for comparison in the experimental evaluation? Including these relevant memory-based approaches would provide a more rigorous validation of MemER's claimed state-of-the-art performance in memory-intensive tasks.

---

> ### Author Response · Authors · 2025-11-19
> **Response to Reviewer JibS (Part 1/2)**
>
> We thank Reviewer JibS for their thoughtful feedback. We have revised the paper **(updates in red text)** and conducted new experiments that we believe address their concerns.
>
> ## Clarification of MemER's Contributions
>
> > The architecture of MemER appears overly simple. The core advancements appear to lie primarily in the training methodology and data preparation for the high-level VLM, and these specific improvements do not seem to be particularly novel. This raises concerns about the overall innovation of the paper.
>
> > Could the authors more clearly articulate the specific innovative contributions of this paper? Given that the architecture is simple and the high-level training methods appear familiar, a clearer distinction from existing work is necessary to establish the technical merit.
>
> Thank you for raising these concerns, and we will provide some elaborations below to address them. We view the simplicity of our approach as a strength, since it makes the system substantially more scalable in practice. Our design is a pragmatic solution that leverages the distinct strengths of both pretrained models: the low-level policy $\pi_{0.5}$ has strong action priors, while Qwen2.5-VL has strong video-understanding priors. The hierarchical split lets the low-level policy remain Markovian (useful for robust retry behavior) and delegates memory and planning to the high-level policy (useful for long-term memory and planning). This modularity is what enables MemER to execute complex, minutes-long tasks from only 50 demonstrations, and our method is directly compatible with existing VLA backbones without architectural changes.
>
> 1. The core novelty is how the high-level policy **interfaces with, updates, and uses memory**. Instead of encoding the entire history or indiscriminately subsampling, MemER maintains a **small set of explicit, informative keyframes** that are (i) nominated online from a bounded context window, (ii) consolidated via a task-agnostic clustering mechanism, and (iii) fed back into the high-level policy at every step. Our ablations show that replacing this memory architecture with standard long-context baselines significantly degrades performance under the same data and optimization setup, indicating that the gains are not due merely to data preparation.
>
> 2. To prove this is not just overfitting via data preparation, we trained a single multi-task policy and added a new cross-task generalization experiment (App F, Fig. 9) to test its ability to handle novel task/object combinations (e.g., using Object Search objects in the Dust & Replace task). Our multi-task model outperforms the single-task models **(82% vs. 59% avg. success)** on these out-of-domain combinations, illustrating its ability to learn a generalizable skill of "what to remember" that transfers to new scenarios, rather than overfitting to the original training demonstrations.
>
> |Setting|Method|Obj. Search|Dust & Replace|Avg|
> |:---|:---|:---:|:---:|:---:|
> |**Standard**|Single-Task|93|86|90|
> ||**Ours (Multi)**|**97**|**96**|**96**|
> |**Cross-Task**|Single-Task|48|70|59|
> ||**Ours (Multi)**|**82**|**83**|**82**|
>
> 3. Because MemER operates by selecting and tracking keyframes from a bounded context window, it **retains salient long-range dependencies while meeting real-time deployment constraints.** Appendix D reports the inference speed and VRAM usage for our high-level policy when conditioned on 0 to 8 keyframes on an NVIDIA GeForce RTX 4090 consumer-grade GPU, copied below. By learning to keep only a few keyframes in context, our method can reason over hundreds of frames while making subsecond subtask predictions.
>
> | Model Configuration | Inference Time (s) | VRAM (GB) |
> | :--- | :---: | :---: |
> | $\pi_{0.5}$ | $0.088 \pm 0.001$ | $6.25$ |
> | MemER (8 recent + 8 keyframes) | $0.787 \pm 0.066$ | $15.93$ |
> | No History (1 recent frame) | $0.532 \pm 0.065$ | $15.55$ |
> | Short History (8 recent frames) | $0.591 \pm 0.064$ | $15.64$ |
> | Long History (32 recent frames) | $0.874 \pm 0.065$ | $16.01$ |
>
> Taken together - (i) the sparse keyframe-based memory, (ii) the multi-task and cross-task generalization results, and (iii) the real-time deployment speed - we believe MemER offers a substantive and novel contribution beyond incremental changes to training or data preparation.

---

> ### Author Response · Authors · 2025-11-19
> **Response to Reviewer JibS (Part 2/2)**
>
> ## Ease of Data Annotation
>
> > The data preparation phase seems to be highly resource-intensive, relying on manual annotation of primitives and the segmentation of trajectories.
>
> Thank you for bringing up this concern, and we will provide some clarifications below to address them.
> - **Subtask annotations and trajectory segmentation:** Generating labels for the language subtasks was a very low-effort process because we can automatically create the list of subtasks based on a randomized high-level task instruction. For instance, for the Counting task, we randomize the # of desired scoops of each ingredient and the corresponding subtasks (i.e. “scoop peanuts into the blue bowl”, “reset scooper”, etc.) can be procedurally generated in order. This is also the case for the Object Search task (randomize which objects are placed in each bin, and which objects we are searching for) and Dust & Replace (randomize which objects are placed on each shelf). During data collection, we just click a button to mark the end of a subtask, and that segment is properly annotated. Additionally, labeling trajectories with language subtasks has become the standard for long-horizon tasks [Hi Robot (Shi et al), DexVLA (Wen et al)]; as such, our framework introduces minimal system complexity and negligible data annotation overhead, allowing it to be automated and agnostic of task, embodiment, and duration.
> - **Keyframe labels:** To get the ground-truth labels for this training, we use a **lightweight, semi-automatic heuristic** that takes only **~5 minutes per task**. We determine a simple annotation rule per subtask - deciding whether or not to keep the last frame of that subtask segment as a ground-truth keyframe, since these transition points usually contain a visually informative state. For example:
>
>     * `look inside <LOCATION> bin` $\rightarrow$ Rule: Save the last frame (to remember what was inside).
>     * `remove object from...` $\rightarrow$ Rule: Save the last frame (to remember the object's original spot).
>     * `reset scooper` $\rightarrow$ Rule: Save no frame (this action is not for memory).
>
>     This simple ruleset is then automatically applied to all 50 demonstrations of each task. We have clarified this process in Sec 3.3 and added the full ruleset to Appendix E.
>
> ## Choice of Baselines
>
> > Why are these models not included as baselines for comparison in the experimental evaluation? Including these relevant memory-based approaches would provide a more rigorous validation of MemER's claimed state-of-the-art performance in memory-intensive tasks.
>
> Thank you for raising this thoughtful suggestion. We have updated the Related Work section with a deeper comparison of MemER against other memory-based approaches. We focused our experimental comparison to “No History”, “Short History”, and “Long History” baselines because these methods capture the memory mechanism used by most related works. Here’s a breakdown of why the specific papers we mentioned in the Related Work were not suitable as direct baselines:
>
> - **SAM2Act** [Fang et al]: This method uses depth information (RGB-D) and 3D reconstruction. MemER uses only RGB images for observations, and it is designed for broad applicability where depth sensors may be noisy or unavailable. Also, SAM2Act’s memory bank operates as a FIFO queue; our “Short History” and “Long History” baselines adequately evaluate this mechanism and demonstrate that it fails to capture minute-long dependencies due to salient information falling out of context.
> - **LiteVLP** [Li et al.]: This method employs a “Multi-Observation Compression” module optimized for fixed-viewpoint settings, where it reduces token counts by filtering static background patches. However, our tasks rely on dynamic wrist-mounted cameras for active perception and precise manipulation. Due to constant ego-motion, the background shifts in every frame, rendering L1-pixel difference compression ineffective. In this setting, LiteVLP reduces to a VLM processing an uncompressed sequence. Thus, our “Long History” baseline (which processes the maximum uncompressed context that is still viable for real-world control) represents the performance upper bound of LiteVLP on our tasks, as it utilizes the same visual history without the risk of motion-induced compression artifacts.
> - **PTP** [Torne et al.]: Past Token Prediction is an auxiliary training objective for diffusion policies, not a distinctive architecture. It is orthogonal to our contribution, as one could train the low-level policy of MemER with the PTP auxiliary objective; as such, we view this as a complementary direction rather than a competing baseline.
>
> By directly comparing against the fundamental **memory mechanism** of these works (FIFO queues, sliding window), we believe we have isolated the specific benefits of MemER’s components.
>
> Thank you for your thorough review. Please let us know if we have adequately addressed your concerns, and we are happy to answer more questions.

---

> > ### Comment · Reviewer_JibS · 2025-11-28
> >
> > Thank you for your explanation and the supplementary experiments. My concerns have been resolved. I am happy to see that the insightful experiments have been incorporated into the revision, and I am particularly interested in the cross-task generalization and data annotation procedure. I will raise my score.

---

### Official Review · Reviewer_bnmy · 2025-11-01

**Soundness:** 3
**Presentation:** 3
**Contribution:** 2
**Rating:** 6
**Confidence:** 4

**Summary:**

- A hierarchical VLA (Vision-Language-Action) framework where the high-level policy retrieves and tracks keyframes from past experience.

- Efficient memory management via online keyframe selection and filtering, reducing redundancy and computational cost.

- Real-world evaluation on three long-horizon tasks requiring minutes of memory: Object Search, Counting Scoops, and Dust & Replace.

**Strengths:**

- Successfully tackles real-world robotic tasks that require reasoning over several minutes of past experience (hundreds of frames), a significant step beyond prior work limited to a few dozen frames.
- The hierarchical design with intelligent keyframe selection avoids the high cost of processing long, raw video sequences, enabling low-latency inference (~1 Hz for high-level policy) suitable for closed-loop control.

**Weaknesses:**

- The framework was evaluated on a single robot arm and on memory within a single task. Its scalability to mobile manipulation, multi-room navigation, and cross-task memory recall remains unexplored.
- The approach is inherently limited to the specific task it was fine-tuned on and lacks the capacity for broader scaling.

**Questions:**

- How does the keyframe extraction algorithm proposed in this work compare to frequency domain-based clustering methods (such as Fourier transform or wavelet transform followed by clustering), for example, with *UniDomain*?
- As tasks become longer, does the computational overhead of maintaining an explicit visual memory buffer adversely impact the performance of the high-level policy by reducing its inference frequency?
- Does the approach presented in this paper offer significant advantages compared to existing long-context strategies, such as introducing sink tokens?

---

> ### Author Response · Authors · 2025-11-19
> **Response to Reviewer bnmy (Part 1/2)**
>
> We thank the reviewer for their valuable feedback and have revised the paper accordingly **(updates in red)**, including additional experiments that we believe address these concerns.
>
> ## MemER is Platform-Agnostic
>
> > The framework was evaluated on a single robot arm and on memory within a single task. Its scalability to mobile manipulation, multi-room navigation, and cross-task memory recall remains unexplored.
>
> Thank you for raising this key point. Although we only tested on a single embodiment, our focus was to explore long-horizon tasks that require remembering long-range temporal dependencies, which we hadn't seen yet in prior work. However, our method of constructing a sparse, persistent memory is completely **platform-independent**, as our keyframe filter is compatible with any stream of image observations. This facilitates extensions with different embodiments beyond a single robot arm (such as mobile manipulators) and different complex, multi-step tasks (such as multi-room navigation), which we are very excited to explore.
>
> To address your concern about single task evaluation, we have run new experiments replacing our single-task models with a **single, multi-task high-level policy** (updated details in Sec 3.3). Our method still significantly outperforms all baselines on the three long-horizon memory-based tasks, achieving **>95%** progress on all tasks, which is on par with an oracle high-level policy.
>
> ## New Generalization Experiments with Multi-Task Model
>
> > The approach is inherently limited to the specific task it was fine-tuned on and lacks the capacity for broader scaling.
>
> We present new results above where we train a single policy on all three tasks, with results demonstrating strong multi-task generalization (Fig. 4). We then evaluated the single- and multi-task models on a new cross-task generalization experiment **(App F, Fig. 9)**, testing its ability to handle novel task/object combinations (e.g., using Search objects in the Dust & Replace task). Our multi-task model outperforms the single-task models **(82% vs. 59% avg. success)** on these out-of-domain combinations, illustrating its ability to learn a generalizable skill of "what to remember" that transfers to new scenarios, rather than overfitting to the original training demonstrations.
>
> |Setting|Method|Obj. Search|Dust & Replace|Avg|
> |:---|:---|:---:|:---:|:---:|
> |**Standard**|Single-Task|93|86|90|
> ||**Ours (Multi)**|**97**|**96**|**96**|
> |**Cross-Task**|Single-Task|48|70|59|
> ||**Ours (Multi)**|**82**|**83**|**82**|
>
> ## Comparison to Frequency Domain-Based Clustering
>
> > How does the keyframe extraction algorithm proposed in this work compare to frequency domain-based clustering methods (such as Fourier transform or wavelet transform followed by clustering), for example, with UniDomain?
>
> This is a great point - we actually explored these heuristic-based methods for selecting keyframes originally, but found them to be much less reliable than semantically selected keyframes. Specifically, MemER’s keyframes come from a trained high-level policy that nominates task-relevant frames (e.g., informative bin views, “dusting complete” states), and the clustering step consolidates these into a compact memory. In contrast, frequency-domain methods (and UniDomain’s energy baseline) operate on low-level intensity/spectral changes, so they mostly detect visual motion, rather than subtask boundaries that correspond to meaningful visual states. Because the high-level policy sees language and multi-view context, it learns to ignore viewpoint noise and only nominate genuinely informative frames. Frequency-domain changes spike on any large camera motion or background shift, so they over-trigger on irrelevant movement in egocentric manipulation.
>
> To illustrate this, we ran UniDomain’s clustering method using energy-based extrema in a sliding window, and observed that the selected keyframes are much less informative compared to MemER's. We take a random Counting demonstration as an example and add the plots to **Appendix I.** To construct a memory buffer that is the same size as our method, the window size had to be >350, meaning that a lot of salient information is lost if it occurs within 350 frames of a local min/max (Figure 12). If we allow the window to be smaller but still sparse enough such that the high-level policy runs at 1Hz (up to 32 frames of memory), we get incredibly noisy and uninformative keyframes **(Figure 13)**.

---

> ### Author Response · Authors · 2025-11-19
> **Response to Reviewer bnmy (Part 2/2)**
>
> ## Compute Analysis
>
> > As tasks become longer, does the computational overhead of maintaining an explicit visual memory buffer adversely impact the performance of the high-level policy by reducing its inference frequency?
>
> We have added new benchmark results to Appendix D detailing the inference speed and VRAM usage for our high-level policy when conditioned on 0 to 8 keyframes on an NVIDIA GeForce RTX 4090 consumer-grade GPU.
>
> | Model Configuration | Inference Time (s) | VRAM (GB) |
> | :--- | :---: | :---: |
> | $\pi_{0.5}$ | $0.088 \pm 0.001$ | $6.25$ |
> | MemER (8 recent + 8 keyframes) | $0.787 \pm 0.066$ | $15.93$ |
> | No History (1 recent frame) | $0.532 \pm 0.065$ | $15.55$ |
> | Short History (8 recent frames) | $0.591 \pm 0.064$ | $15.64$ |
> | Long History (32 recent frames) | $0.874 \pm 0.065$ | $16.01$ |
>
> Since our method operates by selecting and tracking keyframes from a bounded window of context, the computational overhead of constructing a memory buffer does not grow as the tasks become longer, and the inference speed is well under the 1Hz limit to allow for smooth coordination with the low-level controller.
>
> ## Comparison to Alternate Long-Context Strategies
>
> > Does the approach presented in this paper offer significant advantages compared to existing long-context strategies, such as introducing sink tokens?
>
> Our method and traditional LLM sink tokens target quite different issues, so they’re more complementary than competing. Attention sinks are architecture-level tricks for 1D token streams, not a semantic memory system. For long-horizon robot control with high-dimensional visual input, there are some clear advantages of MemER over sink tokens:
>
> - Sink tokens are a tiny set of fixed positions that store an implicit global summary learned during pretraining; you don’t explicitly control which events get preserved. MemER’s high-level policy nominates frames that **actually matter** for the task (e.g. “when did I last see the ketchup?”, “which shelf did I just dust?”). Ground-truth keyframes come from subtask boundaries, so the memory is aligned with task semantics, not just positional quirks.
>
> - Robot tasks here are deeply partially observable. The robot might see an object once, then not see it again for minutes; the keyframe must carry that snapshot forward. MemER’s design stores memory as episodic visual scenes tied to subtasks, not just global text statistics. Attention sinks give you a strong global text prior, but they don’t give you **structured multi-view visual memory** over a changing 3D scene.
>
> Furthermore, recent multimodal work shows that visual attention sinks exist in VLMs as tokens with huge activations (“sink dimensions”) that live in background regions and get a lot of attention but contribute little useful information. This actually tends to be harmful as it results in wasted attention budget and leads to more hallucination and weaker grounding. A bunch of methods explicitly try to counteract this: Visual Attention Redistribution redistributes attention away from visual sinks to more informative tokens to reduce hallucination [1] and Visual Tokens Withdrawal prunes uninformative visual tokens to speed up multimodal inference [2]. For VLMs especially, sink tokens are actually **something to avoid** rather than a mechanism to unlock longer context.
>
> Thank you again for your thoughtful review. We hope our revisions have addressed your concerns, and please feel free to let us know if you have any additional questions or feedback.
>
> **References**
>
> [1] Kang et al. See what you are told: Visual attention sink in large multimodal models, 2025. URL https://arxiv.org/abs/2503.03321.
>
> [2] Lin et al. Boosting multimodal large language models with visual tokens withdrawal for rapid inference, 2025. URL https://arxiv.org/abs/2405.
> 05803.

---

### Official Review · Reviewer_eE1T · 2025-11-02

**Soundness:** 3
**Presentation:** 3
**Contribution:** 2
**Rating:** 4
**Confidence:** 4

**Summary:**

This paper proposes MemER, a hierarchical framework that endows VLAs with long-term memory via experience retrieval. A high-level VLM policy processes recent frames and retrieved keyframes to generate language primitives and nominate new keyframes for storage. A low-level VLA policy executes these primitives based on the current frame. This sparse retrieval mechanism allows the robot to successfully perform complex, multi-minute tasks that depend on recalling distant past events.

**Strengths:**

- The hierarchical design, where a high-level policy learns to explicitly nominate salient keyframes for retrieval, is a novel and highly scalable architecture for managing truly long-horizon (multi-minute) dependencies.
- The experiments are well-designed for long-horizon tasks, and the ablation comparing visual memory ($K_{img}$) to textual memory ($K_{text}$) provides a crucial insight into the limitations of using language as a lossy memory representation.

**Weaknesses:**

- The method's generalizability is unproven, as it was only evaluated on three custom, in-domain tasks and lacks benchmarks on standard, multi-task datasets like RoboCasa or LIBERO.
- The framework introduces significant system complexity and data annotation overhead, as it requires labeling both language primitives and ground-truth keyframes for training the high-level policy.
- The paper provides no computational analysis, making it impossible to assess the inference latency or memory cost of this dual-policy system, which is a critical factor for real-world deployment.
- The comparison to GPT-5 was only conducted offline (Table 2), and it's unclear if this setup accurately reflects the challenges of closed-loop control or simply highlights the need for finetuning.

**Questions:**

- The most important concern is that the dual-policy architecture is computationally heavy and overly complex, requiring two separate, large models (a VLA and a VLM) to be run in parallel. The paper fails to justify this separation or explore a more efficient, unified architecture where a single VLM backbone could perform both high-level planning (keyframe nomination, primitive generation) and low-level feature extraction, which is a significant missed opportunity.

---

> ### Author Response · Authors · 2025-11-19
> **Response to Reviewer eE1T (Part 1/2)**
>
> ## Overview of Main Changes
> We thank the reviewer for their constructive comments. We have revised the paper **(updates in red)** with new:
>
> 1. **Generalization Experiments (Appendix F):** A multi-task policy outperforms single-task baselines (82% vs 59%) on out-of-domain combinations
>
> 2. **Explanation of Annotation Labeling (Sec 3.3, Appendix E)**: Keyframe labeling relies on semi-automatic rules (~5 mins/task), not manual frames.
>
> 3. **Analysis of Compute Usage (Appendix D):** MemER runs at 0.8s / 16GB VRAM on a 4090, meeting 1Hz real-time needs.
>
> 4. **Discussion of Dual-Model Architecture (Appendix J)**: We detail why unified models failed (conflicting priors) vs. our hierarchical approach.
>
> ## Generalization Experiments
>
> > The method's generalizability is unproven, as it was only evaluated on three custom, in-domain tasks and lacks benchmarks on standard, multi-task datasets like RoboCasa or LIBERO.
>
> This is an important concern. Our primary goal is to demonstrate a generalizable framework for tasks requiring **long-horizon memory**, a specific challenge not yet present in standard generalization benchmarks like RoboCasa and LIBERO. To the best of our knowledge, our real-world robotic tasks necessitate reasoning over more image observations than prior work.
>
> **New Generalization Experiments:** To address your concern about in-domain-only evaluation, we ran new experiments that replaced our single-task models with a **single, multi-task high-level policy** (Sec 3.3, Fig. 4). Our method still significantly outperforms all baselines on the three long-horizon memory-based tasks, achieving **>95%** progress on all tasks. We then evaluated the single- and multi-task models on a new cross-task generalization experiment (App F, Fig. 9), testing its ability to handle novel task/object combinations (e.g., using Object Search objects in the Dust & Replace task). Our multi-task model outperforms the single-task models **(82% vs. 59% avg. success)** on these out-of-domain combinations, illustrating its ability to learn a generalizable skill of "what to remember" that transfers to new scenarios, rather than overfitting to the original training demonstrations.
>
> |Setting|Method|Obj. Search|Dust & Replace|Avg|
> |:---|:---|:---:|:---:|:---:|
> |**Standard**|Single-Task|93|86|90|
> ||**Ours (Multi)**|**97**|**96**|**96**|
> |**Cross-Task**|Single-Task|48|70|59|
> ||**Ours (Multi)**|**82**|**83**|**82**|
>
>
> ## Ease of Data Annotation
>
> > The framework introduces significant system complexity and data annotation overhead, as it requires labeling both language primitives and ground-truth keyframes for training the high-level policy.
>
> We clarify these points below:
>
> **Subtask labels:** Generating labels for the language subtasks was a very low-effort process because we can automatically create the list of subtasks based on a randomized high-level task instruction for all our tasks. For instance, for the Counting task, we randomize the # of desired scoops of each ingredient; the corresponding subtasks (i.e. “scoop peanuts into the blue bowl”, “reset scooper”, etc.) can then be procedurally generated in order. During data collection, we just click a button to mark the subtask completion. Additionally, labeling trajectories with language subtasks has become the standard for long-horizon tasks [1, 2]; as such, our framework introduces minimal system complexity and negligible data annotation overhead, allowing it to be automated and agnostic of task, embodiment, and duration.
>
> **Keyframe labels:** To get the ground-truth labels for this training, we use a **lightweight, semi-automatic** heuristic that takes only **~5 minutes per task**. We determine a simple annotation rule per subtask - deciding whether or not to keep the last frame of that subtask segment as a ground-truth keyframe, since these transition points usually contain a visually informative state. For example:
>
> * `look inside <LOCATION> bin` $\rightarrow$ Rule: Save the last frame (to remember what was inside).
> * `remove object from...` $\rightarrow$ Rule: Save the last frame (to remember the object's original spot).
> * `reset scooper` $\rightarrow$ Rule: Save no frame (this action is not for memory).
>
> This simple ruleset is then automatically applied to all 50 demonstrations of each task; this process is not a manual, per-frame effort, but a quick, one-time setup that makes keyframe labeling practically free, compared to the time spent collecting demos, which took ~6 hours per task. We have clarified this in the new “Annotating Keyframes for the High-Level Policy” subsection in Sec 3.3 and added the full ruleset to Appendix E.
>
> [1] Shi et al. Hi robot: Open-ended instruction following with hierarchical vision-language-action models, 2025. URL https://arxiv.org/abs/2502.19417.
>
> [2] Wen et al. Dexvla: Vision-language model with plug-in diffusion expert for general robot control, 2025. URL https://arxiv.org/abs/2502.05855.

---

> ### Author Response · Authors · 2025-11-19
> **Response to Reviewer eE1T (Part 2/2)**
>
> ## Computational Analysis
>
> > The paper provides no computational analysis, making it impossible to assess the inference latency or memory cost of this dual-policy system, which is a critical factor for real-world deployment.
>
> We have added new benchmark results to Appendix D detailing the inference speed and VRAM usage for our high-level policy when conditioned on 0 to 8 keyframes on an NVIDIA GeForce RTX 4090 consumer-grade GPU.
>
> | Model Configuration | Inference Time (s) | VRAM (GB) |
> | :--- | :---: | :---: |
> | $\pi_{0.5}$ | $0.088 \pm 0.001$ | $6.25$ |
> | MemER (8 recent + 8 keyframes) | $0.787 \pm 0.066$ | $15.93$ |
> | No History (1 recent frame) | $0.532 \pm 0.065$ | $15.55$ |
> | Short History (8 recent frames) | $0.591 \pm 0.064$ | $15.64$ |
> | Long History (32 recent frames) | $0.874 \pm 0.065$ | $16.01$ |
>
> **High-Level Policy Inference:** We finetune Qwen2.5VL-7B-Instruct as our new high-level policy, and the subtask prediction is **~0.8s** when there are 8 keyframes + 8 recent frames (16 in total) in the context, which is within our 1Hz threshold for real world deployment. The VRAM usage is **~16GB**, which is within the 24GB limit of a 4090, making it deployable on a single card.
> Our new analysis also shows that if we were to naively increase the context length to 128 frames, the inference time for a subtask prediction becomes **~4s**, which is too slow for real-time deployment. Since our method operates by selecting and tracking keyframes from a bounded window of context, we are able to retain salient long-range dependencies (spanning several hundreds of frames) while maintaining inference speeds critical for real-world deployment.
>
> **Low-Level Policy Inference:** The inference speed of pi0.5 is **~0.09s**, and the VRAM usage is **6.25GB**, which also fits within one 4090.
>
> Together, we are able to comfortably run our entire hierarchical policy (low- and high-level) with just **2 4090s**.
>
> >The comparison to GPT-5 was only conducted offline (Table 2), and it's unclear if this setup accurately reflects the challenges of closed-loop control or simply highlights the need for finetuning.
>
> The reviewer correctly raises two limitations with using frontier VLMs: latency and accuracy. We attempted closed-loop control with GPT-5 and Gemini Robotics-ER 1.5, but they resulted in **0% success** due to the high API latency **(10-15s)**, which caused the high- and low-level policy to be out of sync. To isolate prediction accuracy from latency constraints, we crafted offline evaluations from data collected on actual rollouts of the low-level policy commanded by ground-truth subtasks. MemER outperforms frontier models on these offline evaluations **(avg. trajectory accuracy: 0.78 for MemER vs. 0.42 for GPT-5, 0.18 for Gemini Robotics-ER 1.5)**, highlighting the necessity of finetuning for robot-specific visual memory.  We report the prediction accuracy across the full trajectory and at critical transition points for GPT-5 (and added Gemini Robotics–ER 1.5 given its robotics-specific agentic capabilities) in Table 2.
>
> ## Justification for Two-Model Architecture
>
> > The paper fails to justify this separation or explore a more efficient, unified architecture where a single VLM backbone could perform both high-level planning (keyframe nomination, primitive generation) and low-level feature extraction, which is a significant missed opportunity.
>
> This is an important architectural concern. We originally explored the unified architecture (detailed in Appendix J) but found the hierarchical approach superior for two key reasons:
> - **Conflicting Priors:** Current foundation models are still quite specialized in the context of robotics. $\pi_{0.5}$ has a strong action prediction prior but weak long-horizon video understanding; Qwen2.5-VL has the inverse problem. We found that fine-tuning a single model (either $\pi_{0.5}$ or Qwen2.5VL) to handle both high-frequency motor control (15Hz) and long-horizon memory reasoning (1Hz) from a small dataset of 50 demos leads to unstable training, causing a unified model to fail at both goals.
> - **Sample Efficiency:** By decoupling the policy into a hierarchy, we leverage the pretrained strengths of each model. This modularity allows MemER to learn complex, long-horizon tasks that require multi-minute memory with just 50 demos. The unified model would require orders of magnitude more data and computational resources.
>
> The unified model provides a compelling long-term goal for memory-based VLAs as we scale both pretraining and data collection. The two-model approach for MemER is efficient and pragmatic, as it takes advantage of the strong priors in existing open-source video understanding and robot foundation models.
>
> Thank you for your detailed review. Please let us know if we have adequately addressed your concerns, and we are happy to answer more questions.

---

> > ### Comment · Reviewer_eE1T · 2025-11-25
> >
> > I thank the authors for their comprehensive rebuttal. The new generalization experiments, computational analysis , and clarification on annotation costs adequately address my concerns. I appreciate the honest acknowledgment of the dual-policy architecture's limitations and the reasonable justification based on conflicting priors in current foundation models—a challenge familiar to VLA researchers.
> >
> >
> > Considering both the paper's improvements and the fundamental limitations of current VLA paradigms, I decide to raise my score to 6. While I recognize this as a pragmatic solution given the current state of foundation models, I strongly encourage exploring unified architectures in future work, as this remains an important open challenge for the field.

---

### Author Response · Authors · 2025-11-24
**General Response**

## Overall Positive Reaction to Rebuttals Prior to Score Reversal
We appreciate all of the reviewers’ constructive responses, which shaped our new experiments and updated manuscript. Among the reviewers who responded to our rebuttals (**eE1T**, **JibS**, and **uYHS**), all expressed a positive reaction and confirmed **raising their initial scores**. In brief:


### From eE1T (4 -> 6):
> Considering both the paper's improvements and the fundamental limitations of current VLA paradigms, I decide to raise my score to 6.


### From JibS (4 -> 6):
>  My concerns have been resolved. I am happy to see that the insightful experiments have been incorporated into the revision, and I am particularly interested in the cross-task generalization and data annotation procedure. I will raise my score.


### From uYHS (6 -> 8):
> The evaluation of scalability aspects, such as memory size and retrieval latency, as well as keyframe generation, has already been addressed. The authors clearly conveyed this in the paper, and it is done very well.

and

> Thank you for the authors’ response. I believe the proposed method is a valuable attempt at enhancing hierarchical VLA with an explicit memory module.


## Initial Review Highlights
We are grateful for all the reviewers’ initial feedback and valuable suggestions. We particularly appreciate the acknowledgment of:
- Simplicity (**JibS**, **uYHS**) and scalability (**eE1T**, **uYHS**) of our method
- Effectiveness for long-horizion tasks requiring multi-minute memory (**bnmy**, **JibS**, **uYHS**)
- Technological contribution in hierarchical policy with intelligent keyframe selection (**eE1T**, **bnmy**)
- Experimental validation through real-world memory-aware tasks with long-range dependencies (**eE1T**, **JibS**, **bnmy**, **uYHS**)
- Clear presentation and motivation (**JibS**, **uYHS**)

## Main Concerns Addressed
- Generalizability (**eE1T**, **uYHS**): We replace our single-task models with a single, multi-task high-level policy which maintains significant improvement over all baselines. We then evaluate on novel task/object settings, where our multi-task model substantially outperforms the single-task models.
- Compute Analysis (**eE1T**, **bnmy**, **uYHS**): We benchmarked our inference and memory requirements on a consumer-grade 4090 GPU, showing our method can reason over hundreds of frames while making subsecond (~0.8s) subtask predictions.
- Cost of Data Annotations (**eE1T**, **JibS**): We clarify the low-effort process of automatically generating subtask annotations and the lightweight review (~5 minutes per task) required to identify ground-truth keyframes.

We address these key concerns and provide new experimental results that strengthen our paper’s contributions. We include our clarifications and additional analyses in our revised manuscript.
## Summary of Updates
New Experiment Results:
- New results from our single, multi-task policy in place of our single-task models (**Figure 4, Table 1**)
- Out-of-domain cross-task results to further demonstrate generalization (**Appendix F**)
- New benchmark results detailing the inference speed and VRAM usage for our high- and low-level policy (**Appendix D**)
- New Gemini Robotics-ER 1.5 comparison (**Section 4.1, Table 2**)

Manuscript Updates:
- Enhanced related work discussion with newly suggested references (**Section 2**)
- Clarified the lightweight process of ground-truth keyframe annotation (**Section 3.3**)
- Specific annotation rules for keyframe labeling (**Appendix E**)

We have highlighted all updates in red for easy reference. We believe MemER is a valuable step towards solving long-horizon tasks that require multi-minute memory.

We sincerely appreciate the reviewers for their insightful comments during the discussion period. We hope our responses have fully addressed all concerns and welcome any further discussion.

---

### Meta-Review · Area_Chair_Qht4 · 2026-01-05

**Summary:**

The submission presents MemER, a simple method for helping VLAs handle long-horizon tasks by having a finetuned VLM keep track of important keyframes from the executed part of the trajectory, "recall" them as appropriate, and use them when issuing instructions to the VLA.

The initial major concerns of the reviewers were as follows:
- Generalizability: evaluation was done on separate tasks featured only in this paper and also omitted mobile manipulation and navigation scenarios.
- Cost and complexity of producing the required training annotations identifying the important keyframes.
- Computational overhead of the proposed method.
- Ostensible simplicity of the proposed method, calling into question its novelty.
- Lack of comparisons to baseline methods for long-horizon decision-making.

The authors provided rebuttals, and three out of four reviewers explicitly acknowledged in their rebuttal responses that the authors managed to mostly address these concerns.

Having read the submission, the metareviewer thinks that the method's simplicity is actually its major strength, coupled with its effectiveness. and in no way detracts from its novelty. The rebuttal experiments have added significant value to the submission's initial analysis. Since the reviewers, too, found their concerns allayed, and since the proposed method is simple to apply for an important problem that so far has no widely accepted solution in robot learning, the metareviewer recommends acceptance.

**Reviewer Concerns:**

The concerns listed above have been mostly addressed, as acknowledged by three out of four reviewers, and they overlap with the concerns of the only reviewer who didn't respond. The only aspect the authors didn't address is experiments on mobile robots, but the metareviewer doesn't view them as necessary for making a case for the MemER's efficacy.

**Reviewer Scores:**

- eE1T mentioned they would increase their score from 4 to 6.
- JibS promised to increase their score up from 4. The authors claim that the score was increased to 6. This is believable, since the reviewer acknowledged that their concerns were addressed, but in any case it would almost surely go up at least to 5.
- uYHS didn't mention changing their score up from 6, although the authors state that it was increased to 8. As above, the metareviewer is willing to believe that the score would be increased, since the reviewer's concerns were addressed.
- bnmy didn't respond, but their initial score was 6 and their concerns were mostly addressed too, so their score would stay the same or perhaps go up to 7.

---

### Decision · Program_Chairs · 2026-01-26

Accept (Poster)